# EAE of Mice: Enzymatic Cross Site-Specific Hydrolysis of H2A Histone by IgGs against H2A, H1, H2B, H3, and H4 Histones and Myelin Basic Protein

**DOI:** 10.3390/ijms24108636

**Published:** 2023-05-12

**Authors:** Andrey E. Urusov, Kseniya S. Aulova, Pavel S. Dmitrenok, Valentina N. Buneva, Georgy A. Nevinsky

**Affiliations:** 1Institute of Chemical Biology and Fundamental Medicine of the Siberian Division of Russian Academy of Sciences, Novosibirsk 630090, Russia; 2G. B. Elyakov Pacific Institute of Bioorganic Chemistry, Far East Division, Russian Academy of Sciences, Vladivostok 690022, Russia

**Keywords:** C57BL/6 mice, EAE model of human multiple sclerosis, immunization mice with MOG and DNA–histone complex, catalytic antibodies, hydrolysis of histones and myelin basic protein, cross-complexation and catalytic cross-reactivity

## Abstract

Histones play vital roles in chromatin function and gene transcription; however, they are very harmful in the intercellular space because they stimulate systemic inflammatory and toxic responses. Myelin basic protein (MBP) is the major protein of the axon myelin–proteolipid sheath. Antibodies–abzymes with various catalytic activities are specific features of some autoimmune diseases. IgGs against individual histones (H2A, H1, H2B, H3, and H4) and MBP were isolated from the blood of experimental-autoimmune-encephalomyelitis-prone C57BL/6 mice by several affinity chromatographies. These Abs–abzymes corresponded to various stages of EAE development: spontaneous EAE, MOG, and DNA–histones accelerated the onset, acute, and remission stages. IgGs-abzymes against MBP and five individual histones showed unusual polyreactivity in the complex formation and enzymatic cross-reactivity in the specific hydrolysis of the H2A histone. All the IgGs of 3-month-old mice (zero time) against MBP and individual histones demonstrated from 4 to 35 different H2A hydrolysis sites. The spontaneous development of EAE over 60 days led to a significant change in the type and number of H2A histone hydrolysis sites by IgGs against five histones and MBP. Mice treatment with MOG and the DNA–histone complex changed the type and number of H2A hydrolysis sites compared to zero time. The minimum number (4) of different H2A hydrolysis sites was found for IgGs against H2A (zero time), while the maximum (35) for anti-H2B IgGs (60 days after mice treatment with DNA–histone complex). Overall, it was first demonstrated that at different stages of EAE evolution, IgGs–abzymes against individual histones and MBP could significantly differ in the number and type of specific sites of H2A hydrolysis. The possible reasons for the catalytic cross-reactivity and great differences in the number and type of histone H2A cleavage sites were analyzed.

## 1. Introduction

Histones and their various modified forms play a vital role in chromatin function. Free extracellular histones usually act as damage factors [1]. The treatment of experimental mice with five exogenous histones adduces to systemic toxic responses, consequently inflammatory reactions and the activation of toll-like receptors [1]. The treatment of mice with antibodies (Abs) neutralizing five histones, heparin, activated protein C, and thrombomodulin provides mice protection against sepsis, trauma, lethal endotoxemia, ischemia–reperfusion injury, pancreatitis, stroke, peritonitis, coagulation, and thrombosis. The increase in the blood’s free histones and nucleosome fragments causes several pathophysiological processes, including the progression of inflammatory processes, several autoimmune diseases (AIDs), and cancer [1].

The core tetramers of nucleosome particles contain two molecules of H4 and H3 and are circled by two dimers of histones H2A and H2B bound with two supercoiled turns of double-stranded DNA [2]. The H1 histone is essential for packing chromatin and forming a higher-order chromatin structure. H2A can also differentiate or regulate chromatin.

In several different AIDs, including multiple sclerosis (MS) and systemic lupus erythematosus (SLE), Abs for DNA and histones are primarily directed against DNA–histone nucleosomal complexes emerging in the blood due to the apoptosis of cells [3].

MS is an inflammatory demyelinating AID of the central nervous system. Many macrophages characterize this pathology, including T-lymphocytes in the perivascular infiltrate [4]. Some data attest to MS pathogenesis due to the essential role of B-cells and auto-Abs against myelin auto-antigens, including myelin basic protein (MBP) [4,5,6]. The activated myelin-reactive CD4+ T-cells could be the principal mediators of multiple sclerosis [4].

Several experimental autoimmune encephalomyelitis (EAE) mice models well mimic the peculiar properties of human MS (for review, see [7,8]). Autoimmune diseases were first supposed to originate from the particular defects of bone marrow hematopoietic stem cells (HSCs) [9]. Later, it was demonstrated that the spontaneous and antigen-induced evolution of autoimmune diseases develops from a specific autoimmune reorganization of bone marrow HSCs [10,11,12,13,14,15,16]. C57BL/6 mice prone to EAE were used earlier for the investigation of the possible mechanisms of spontaneous and accelerated development of EAE with myelin oligodendrocyte glycoprotein (MOG) [12,13] as well as DNA–protein complexes [10,11]. It showed that the treatment of SLE-prone MRL-lpr/lpr mice with DNA–protein complexes [14,15,16] and C57BL/6 mice with DNA–histone complexes or MOG [10,11,12,13] causes a significant speed-up of SLE and EAE development. Such acceleration is a consequence of specific changes in HSC differentiation profiles and a significant increase in lymphocyte proliferation in various organs of mice [10,11,12,13,14,15,16]. Moreover, these changes in the differentiation profiles are bound to produce various auto-Abs–abzymes (Abzs) that hydrolyze DNA, polysaccharides, RNA, proteins, and peptides. The detection of different auto-abzymes are the earliest and statistically significant markers of many AID onsets and progressions [10,11,12,13,14,15,16,17,18,19,20,21,22,23]. The catalytic activities of Abzs are easily detectable even at the onset of several AIDs (at the pre-disease stage corresponding to the beginning of AIDs) before the discovery of typical markers of different autoimmune pathologies [12,13,14,15,16,17,18,19,20,21,22,23]. The titers of auto-Abs for peculiar auto-antigens at the onset of many AIDs usually correspond to standard indices of detection ranges in the blood of conditionally healthy humans and mice. The appearance of plural Abzs clearly demonstrates the start of autoimmune reactions when an increase in the catalytic activities of antibodies is linked to the development of profound pathologies. However, several parallel mechanisms can provide different autoimmune pathology developments, eventually resulting in a breakdown of self-tolerance [20,21,22,23].

Natural auto-abzymes degrading various oligosaccharides, proteins, peptides, nucleotides, DNA, and RNA were detected in the blood of patients with several AIDs and viral diseases [18,19,20,21,22,23]. Auto-abzymes with deficient activities hydrolyzing polysaccharides [24], thyroglobulin [25,26], and vasoactive neuropeptide [27] were found in the blood of some conditionally healthy volunteers. However, conditionally healthy humans and animals usually lack Abzs [18,19,20,21,22,23]. Moreover, some germline auto-antibodies of healthy people can possess detectable levels of some amyloid-, superantigen-, and microbe-directed activities [25,28].

Similar to SLE patients [22], the blood plasma of MS patients contains abzymes splitting DNA and RNA [29,30,31], MBP [32,33,34,35], oligosaccharides [20,21,22,23], and histones [36]. The relative activities (RAs) of IgGs–Abzs from the cerebrospinal fluids of patients with MS degrading MBP, polysaccharides, and DNA are, on average, 30–60 times higher than from the blood of the same patients [37,38,39]. In various AIDs, abzymes against MBP can hydrolyze MBP in the myelin–proteolipid sheath of axons and may, therefore, have a very adverse role in MS, SLE, and other AID pathogeneses [18,19,20,21,22,23].

Abs–abzymes splitting five histones (H1, H2A, H2B, H3, and H4) were found in the sera of MS [36], HIV-infected patients [40,41,42,43,44,45], and EAE mice [11,46]. As mentioned above, the five free extracellular histones act as damage molecules [1]. DNA–histone complexes are the momentous auto-antigens producing Abs and abzymes against DNA and histones [4], which are very toxic for mammals. These abzymes can penetrate through membranes of cells and nuclei, hydrolyze chromatin DNA, and induce the apoptosis of cells [47,48,49]. Therefore, Abzs splitting MBP, DNA, and histones may be essential in the pathogenesis of MS and other AIDs.

It is believed that the development of different AIDs could be associated with human infection by various bacteria and/or viruses, including human herpesvirus, human endogenous retroviruses, and Epstein–Barr virus (for review, see [50,51,52]). At first, there might be the production of Abs against bacterial or viral compounds, which possess a high level of homology with human proteins [53,54,55]. Later, due to the strong mimicry of some viral or bacterial proteins with those of humans, immune system violations can result in the generation of auto-Abs for human proteins and the evolution of AIDs. The treatment of different autoimmune-prone mice with different antigens leads to a significantly higher incidence of abzyme production with higher enzymatic activities than in the normal conventionally used mouse strains [56,57]. 

The unspecific complex formation of various proteins and enzymes with foreign ligands is described as a widespread phenomenon [58,59,60]. The efficiency of the correct selection of specific substrates by enzymes during the complex formation is usually at most 1–2 orders of magnitude [58,59,60]. Subsequent specific changes in enzyme and substrate structures lead to the catalysis stage, and to an increase in the reaction rate by 5–8 orders of magnitude for specific in comparison with non-specific substrates [58,59,60]. Therefore, enzymatic cross-reactivity regarding substrates in the case of canonical enzymes is a very sporadic case [58,59,60]. Typically, classical enzymes usually catalyze only one chemical reaction.

The non-specific complex formation of some proteins, nucleic acids, and other ligands with antibodies against others detectable by the enzyme-linked immunosorbent assay (ELISA) or affinity chromatography is a widespread phenomenon known as Abs polyspecificity or polyreactivity complexation [61,62,63,64]. Abzymes against many different proteins, similar to classical enzymes, usually specifically hydrolyze only one specific antigen–protein and cannot split many other control unspecific globular proteins ([18,19,20,21,22,23] and refs therein). First, it was shown that anti-MBP Abzs hydrolyze only MBP [32,33,34,35,36], while abzymes against histones hydrolyze only histones [40,41,42,43]. However, an analysis of abzymes showed that the immune response to auto-antigens in AIDs is much more complex and multifaceted than can be seen based on classical immunology.

The catalytic cross-activity of any Abs–Abzs against various proteins was just recently described [18,19,20,21,22,23]. It was shown that auto-IgGs of HIV-infected patients against MBP specifically hydrolyze MBP and the five H1–H4 histones and vice versa; total preparations of abzymes against the five histones effectively degrade MBP [44,45]. Recently, IgGs against five histones and MBP corresponding to different spontaneous, MOG, and DNA–histones that accelerated the onset, acute, and remission stages of C57BL/6 mice EAE development were analyzed [46]. Such IgGs–abzymes against five histones and MBP demonstrated unusual polyreactivity in complex formation and enzymatic cross-reactivity in the hydrolysis of the H4 histone. These chimeric Abs with cross-catalytic reactivity can be very hazardous for developing many AIDs since Abzs against the five histones can hydrolyze MBP of nerve tissue shells. It is interesting to what extent the phenomenon of the enzymatic cross-reactivity between Abzs against histones and MBP is common for humans and animals with different AIDs. In addition, it is important to understand whether there is an unusual enzymatic cross-reactivity of Abs–Abzs against histones and MBP only in the case of H4, or also for other histones. In addition, the analysis of abzymes corresponding to different stages of EAE development by C57BL/6 mice allows an understanding of how the relative activities of abzymes and their substrate specificity concerning individual histones and MBP can change depending on the stage of pathology.

Here, it was shown that the abzymes of mice against five individual histones and MBP possess catalytic cross-reactivity in the splitting of the H2A histone. Moreover, it was demonstrated that abzymes against each of the five histones and MBP corresponding to different stages of EAE development can hydrolyze the H2A histone with different efficiencies and at various specific sites. 

## 2. Results

### 2.1. Choosing a Model for Catalytic Cross-Reactivity Analysis

Theoretically, the human immune system can produce ~10^6^ variants of Abs against one antigen having different properties [65]. The possibilities of ELISA and affinity chromatography in studying the possible diversity of antibodies against external and internal specific antigens in the blood of conditionally healthy donors and patients with AIDs are very limited.

As shown in many studies (for review, see [18,19,20,21,22,23]), only the analysis of abzymes can reveal an exceptionally expanded diversity of antibodies against the same antigens since, unlike antibodies without activity, abzymes differ not only in their affinity for the substrate but also in the rate of hydrolysis, optimal pH, dependence or independence from various ions of one and two valence metals, etc. An approximate evaluation of the possible diversity of abzymes in terms of enzymatic properties for one antigen was carried out in several works [66,67,68,69,70,71,72]. A cDNA library of light chains (κ-type) of Abs from SLE patients and a phage display method were used to obtain individual monoclonal light chains (MLChs) of Abs. The pool of phage particles was divided into ten peaks eluted from MBP–Sepharose by different NaCl concentrations. MLChs bound to phage particles of all 10 peaks eluted from affinity sorbent effectively hydrolyzed MBP. The phage particles of one peak eluted with 0.5 M NaCl were used to obtain individual colonies and isolation MLChs corresponding to them [66,67,68,69,70,71,72]. MLChs corresponding to 72 randomly selected of 440 individual colonies were analyzed, and 22 of them (~30%) possessed MBP-hydrolyzing activity. Of the 22 MLChs with a comparable affinity for MBP, 12 possessed metalloprotease, four were serine-like, and three were thiol proteases. Two MLChs demonstrated serine-like and metalloprotease activities combined in one active site, and one had three activities: serine-like, metalloprotease, and DNase activity [66,67,68,69,70,71,72].

It is important that, unlike other methods of antibody analysis, the analysis of the optimal conditions for the manifestation of catalytic activity allows us to distinguish between antibodies with comparable affinities for the same antigen. It showed that all preparations of MLChs–abzymes differed greatly in relative activity, optimal concentrations of different metal ions (Na^+^, K^+^, Mg^2+^, Zn^2+^, Mn^2+^, Co^2+^, Ni^2+^, etc.), as well as pH optima [66,67,68,69,70,71,72].

A homology analysis of the MLCh protein sequences of several classical Zn^2+^- and Ca^2+^-dependent, as well as human serine-like and thiol proteases, was carried out. The DNA sequences of these MLChs were analyzed, and they were identical (88–100%) to the germ lines of the IgLV8 light-chain genes of several described antibodies [70,71,72]. The MLCh protein sequences responsible for binding MBP, metal ion chelation, and direct catalysis turned out to be very close to those for canonical proteases [70,71,72].

It should be emphasized that antibodies of all the peaks eluted from MBP–Sepharose possessed MBP-hydrolyzing activities. If we take into account the average value of the percentage of active abzymes in one peak, which is approximately 30–35%, and the minimum peak value of 10, as well as the number of individual colonies analyzed, then the possible number of Abzs of only κ-type with MBP-hydrolyzing activity in SLE patients can be ≥1000. However, the MBP-hydrolyzing activity was shown to possess Abs with kappa and lambda chains [20,21,22,23]. These data testify to the extreme diversity of abzymes for the same antigen. In addition, it was evident that in the same active center of abzymes, unlike classical enzymes, amino acid residues responsible for the manifestation of several catalytic functions by abzymes can be combined.

The evolution of EAE in C57BL/6 mice occurs spontaneously. The treatment of mice with DNA–histone complexes [11,12] or MOG can significantly accelerate EAE development [13,14]. There are three main stages of EAE progression after mice immunization with the antigens: the onset at 7–8, the acute at 18–20, and the remission phase at 25–30 days. The acceleration of EAE development is associated with specific changes in bone marrow HSC differentiation profiles and the rise in lymphocyte proliferation [11,12,13,14]. These processes stimulate the production of lymphocytes synthesizing Abzs, splitting DNA, RNA, MBP, MOG, and histones. The parameters characterizing all these specific changes in mice were analyzed earlier [10,11,12,13,14,15,16]. 

To assay the enzymatic cross-reactivity of IgGs in the hydrolysis of H2A histone, we chose two antigens described earlier: MOG [13,14] and the DNA–histone complex [10,11]. Data showing the changes in the differentiation profile of HSCs before and after mice immunization with MOG and the DNA–histone complex are presented in the Appendix A. The changes in the relative concentrations of different antibodies are given in Appendix A. The changes in the relative activities of IgGs–abzymes in the hydrolysis of MOG, MBP, DNA, and histones during the EAE development are given in Appendix A. One can see that during the development of spontaneous EAE, the rise in the relative amounts of four precursors of hemopoietic cells (CFU-E, CFU-GM, BFU-E, and CFU-GEMM) in the bone marrow of C57BL/6 mice are relatively gradual and slow. Mice treatment with MOG and the DNA–histone complex results in various changes over time in the profile of stem cell differentiation, but EAE development accelerates in all cases.

Recently, we analyzed a possible enzymatic cross-reactivity of the total IgGs of C57BL/6 mice against five histones (isolated using Sepharose containing five immobilized histones) and MBP in the hydrolysis of the H4 histone [46]. It showed that the treatment of mice with MOG and the DNA–histone complex results in the hydrolysis of the H4 histones in different sites. In addition, the sites of H4 histone hydrolysis by IgGs against the five histones and MBP significantly change depending on the stage of development of EAE: onset, acute phase, and stage of remission. In this work, a more detailed study was performed in which the catalytic cross-activity of Abs against five individual histones and MBP in H2A histone hydrolysis using IgGs against five H1–H4 histones. IgGs against each of the five individual histones were obtained from the fraction of IgGs against all five histones. The following groups of mice were used to purify the total IgGs against the five individual histones and MBP corresponding to the different stages of EAE (Table 1).

### 2.2. Purification of Antibodies

The purification of electrophoretically homogeneous IgG preparations (IgG_mix_; the mixture of seven plasma blood samples) corresponding to each mice group using Protein G–Sepharose and FPLC gel filtration under drastic conditions (pH 2.6) was described previously [46]. After SDS-PAGE of IgG_mix_ preparations, protease activity in the hydrolysis of histones and MBP was detected only in one 150 kDa IgG band, and there were no other proteins or protease activity peaks (Appendix A). This indicated that the IgG_mix_ preparations corresponding to each of the mice groups do not contain impurities of canonical proteases having molecular weight 20–40 kDa.

IgGs against five histones corresponding to each mice group were purified earlier by chromatography on histone5H–Sepharose containing five immobilized histones [46]. Non-specifically bound and having low affinity for the five histones, IgG fractions were first eluted with 0.2 M NaCl. Anti-histone-specific IgGs having a high affinity for the five histones were then eluted with Tris–Gly buffer, pH 2.6 as in [46]. For the additional purification of IgGs against the five histones from potential impurities of IgGs against MBP, the fraction from histone5H–Sepharose was passed through MBP–Sepharose. The fraction obtained upon loading onto MBP–Sepharose was further used as anti-5-histones IgGs for purification of IgGs against each of the 5 individual histones (Table 1).

The IgG_mix_ fractions eluted upon loading from histone5H–Sepharose was used to isolate anti-MBP IgGs by their chromatography on MBP–Sepharose. IgGs with low affinity for MBP–Sepharose were eluted using NaCl (0.2 M). Anti-MBP Abs were eluted using an acidic buffer (pH 2.6) [46]. For the additional removal of anti-MBP IgGs against potential impurities of anti-histone IgGs, the fraction eluted from MBP–Sepharose was subjected to re-chromatography on histone5H–Sepharose [46]. The fraction of IgGs eluted from the histone5H sorbent at the loading was named anti-MBP IgGs (Table 1). 

### 2.3. SDS-PAGE Analysis of Catalytic Cross-Reactivity

A possible enzymatic cross-reactivity of anti-5-histone IgGs (eluted from histone5–Sepharose) and anti-MBP IgGs (eluted from MBP–Sepharose) was first analyzed using SDS-PAGE. Figure 1A,B demonstrate the hydrolysis of H2A histone with anti-5-histone IgGs and anti-MBP IgGs, while Figure 1C,D shows the hydrolysis of MBP by these IgGs. 

The efficiency of H2A and MBP splitting with various IgGs was calculated from the decrease in the proteins in the initial bands (lane C, Figure 1) after incubation with IgGs compared to their content after incubation without Abs. After 14 h of H2A incubation with IgGs against MBP and histones, the relative content of H2A (~14.0 kDa) and MBP (18.5 kDa) formed decreased remarkably and or significantly compared to the control experiment (lanes C). These data suggest that anti-histone IgGs and anti-MBP of mice possess a known phenomenon of unspecific antibody complex formation polyreactivity [61,62,63,64] and enzymatic cross-reactivity in MBP and histone hydrolysis. These data, however, cannot provide absolute proof of the enzymatic cross-reactivity between IgGs–abzymes against MBP and the five histones because it cannot be ruled out that after their isolation by several affinity chromatographies, the obtained IgGs can contain very small admixtures of alternative IgGs. More powerful evidence of enzymatic cross-reactivity may be obtained from the significant difference in the specific sites of H2A histone hydrolysis by IgGs against MBP and the five histones. This study first analyzed the possibility of the hydrolysis of histone H2A with specific IgGs–abzymes against the five individual histones and MBP. 

### 2.4. MALDI Analysis of H2A Histone Hydrolysis

As shown using IgGs–abzymes of HIV-infected patients against five histones, they hydrolyzed all histones and MBP and vice versa [44,45]. It showed that C57BL/6 mice IgGs also hydrolyzed all five histones and MBP [10,11,12,13]. Moreover, IgGs against MBP and five histones of C57BL/6 mice demonstrated cross-enzymatic activity in H4 histone splitting [46]. In addition, the sites of hydrolysis of H4 by a set of different IgGs mentioned in Table 1 significantly depend on their type and the stage of EAE development.

In this work, first IgGs from mice corresponding to individual histones and MBP and different stages of EAE development (Table 1) were used to study their enzymatic cross-reactivity in the hydrolysis of the H2A histone. The IgGs were used for identifying the cleavage sites of H2A by MALDI mass spectrometry. After the addition of the IgGs (Figure 2A), the H2A histone was almost homogeneous, showing two signals of its one- (*m*/*z* = 13981.9 Da) and two-charged ions (*m*/*z* = 6991 Da). First, H2A hydrolysis sites were determined using IgGs against five individual histones corresponding to the zero time of the experiment (3-month-old mice) and the spontaneous development of EAE over 60 days (without mice immunization). For several IgG preparations against individual histones listed in Table 1, 8–10 spectra were obtained. Several typical spectra are shown in Figure 2. Each preparation of IgGs corresponded to a specific set of peaks corresponding to the hydrolysis of the H2A histone.

Figure 3, Figure 4, Figure 5 and Figure 6 demonstrate spectra corresponding to the hydrolysis of H2A by IgGs against various histones and MBP after the immunization of mice with MOG and the DNA–histone complex.

### 2.5. Sites of H2A Histone Hydrolysis

As can be seen from Figure 2, Figure 3, Figure 4, Figure 5 and Figure 6, each antibody preparation demonstrates its own specific set of peaks corresponding to hydrolysis products with different molecular weights. All sites of H2A histone hydrolysis by all IgGs against individual histones and MBP are summarized in Figure 7.

Some data on the sites of H2A hydrolysis by IgGs against H3 and H4 histones are shown in Figure 8. 

One can see that, overall, the sites of H2A histone cleavage by IgGs against five individual histones H1–H4 and MBP corresponding to the beginning of the experiment (3-month-age mice), 60 days of spontaneous development of EAE, 20 days after mice treatment with MOG, as well as 20 and 60 days after immunization with the DNA–histone complex are substantially or very different and are predominantly located in specific amino acid (AA) clusters of different lengths.

All IgGs hydrolyze H2A histone in many various sites, the number of which also differs for different IgG preparations. For a more straightforward comparison of the different sites of hydrolysis of H2A with IgGs against three histones, they are given in Table 2. 

Antibodies against H2A histone by IgGs of 3-months-old mice (zero time, Con-aH1-0d) hydrolyzed this histone at 7 sites, and after 60 days of spontaneous EAE development (Spont-aH1-60d) at 28 sites; only 5 sites were the same (Table 2). In addition, only one major of the seven sites of H2A hydrolysis by the Spont-aH1-60d (R17-S18) was the same as only one major site for Con-aH1-0d (Table 2). 

During the spontaneous development of EAE, the number of histone H2A hydrolysis sites by Abs against H2A increased from 4 (Con-aH2A-0d) to 17 (Spont-aH2A-60d). Interestingly, in both cases, there was only one major site of H2A hydrolysis, R17-S18, but on the whole, the sites were very different (Table 2). 

The number of sites of H2A hydrolysis by antibodies against H2B histone (Con-aH2B-0d) at zero time was only 8, but increased to 27 sites after 60 days (Spont-aH2B-60d); none of the 4 major sites for Spont-aH2B-60d coincided with one major site for Con-aH2B-0d. 

A somewhat different situation in H2A hydrolysis was observed for IgGs against H3 and H4 histones. The number of H2A hydrolysis sites for Spont-aH3-60d decreased in comparison with Con-aH3-0d from 15 to 12, and in general there were no identical sites (Table 2). 

The number of H2A hydrolysis sites for IgGs against the H4 histone (Con-aH4-0d) decreased from 8 to 7 after 60 days of spontaneous EAE (Table 2); all the sites of hydrolysis were also different (Table 2).

When analyzing these data, it should be taken into account that a change in the differentiation profile of stem cells leads to the fact that the synthesis of Abs by B-lymphocytes already occurs at the level of the cerebrospinal fluid of the bone marrow [37,38,39]. Antibodies isolated from the cerebrospinal fluid of the bone marrow of patients with MS had a 30–60 times higher activity in the hydrolysis of DNA, MBP, and oligosaccharides compared with antibodies from the blood of the same patients [37,38,39]. In addition, at different stages of the spontaneous development of SLE and EAE, several periods of changes in the differentiation profile of bone marrow stem cells were observed [10,11,12,13,14,15,16].

Thus, during the spontaneous development of EAE, there was an expansion or contraction of the repertoire of lymphocytes producing abzymes with completely different properties compared to zero time. Interestingly, the small fractions of pools of antibodies against H1, H2A, and H2B contained IgGs hydrolyzing H2A at the same sites as Abs corresponding to zero time together with small fractions of abzymes splitting histone at completely different sites.

With the spontaneous development of EAE, there was a complete change in the type of lymphocytes synthesizing IgGs against H3 and H4 histones, capable of hydrolyzing the H2A histone. The sites of hydrolysis of H2A by IgGs against H3 and H4 histones at time zero and 60 days are completely different (Table 2).

The immunization of mice with MOG and the DNA–histone complex can lead to a different change in the profile of stem cell differentiation compared to the spontaneous development of EAE and, therefore, the appearance of IgGs against five histones with different catalytic properties in the hydrolysis of H2A histone. As mentioned above, the treatment of C57BL/6 mice with MOG led to a sharp acceleration of the development of EAE up to 20 days, corresponding to the acute phase with subsequent remission of the pathology (>25 days) [10,11,12,13,14]. Table 3 shows data on the sites of hydrolysis of the H2A histone by antibodies against five different histones, corresponding to 20 days after mice immunization with MOG.

At time zero, the number of sites of H2A hydrolysis by IgGs against H1 was seven (Con-aH1-0d; Table 2), and at 20 days after mice immunization with MOG-15 (MOG20-aH1; Table 3). Interestingly, only three sites were common for these IgGs. At 60 days after the spontaneous development of EAE, the site number of H2A hydrolysis with IgGs against H1 was 28 (Spont-aH1-60d; Table 2), and only 8 of them were the same as hydrolysis sites for MOG20-aH1 (Table 3).

The number of sites of H2A hydrolysis by anti-H2A antibodies in zero time (Con-aH2A-0d; Table 2) increased after immunization of mice with MOG (MOG20-aH2A; Table 3) from 4 to 11, and 3 sites out of 4 coincided with those for MOG20-aH2A. Of the 17 sites of H2A hydrolysis by anti-H2A antibodies corresponding to 60 days of spontaneous EAE development (Spont-aH2A-60d; Table 2), 6 sites coincided with those for MOG20-aH2A (Table 3).

The number of sites of H2A hydrolysis by IgGs against H2B histone at time zero was 8 (Con-aH2B-0d; Table 1) and increased to 10 after mice treatment with MOG (MOG20-aH2B; Table 3). In this case, 6 out of 8 sites of H2A hydrolysis by Abs against H2B at zero time (Con-aH2B-0d; Table 1) coincided with 6 out of 10 sites for MOG20-aH2B (Table 3). However, the spontaneous development of EAE led to an increase in H2A hydrolysis sites by anti-H2B IgGs (Spont-aH2B-60d; Table 1) up to 27. Of the 27 sites corresponding to the spontaneous development of EAE within 60 days (Spont-aH2B-60d; Table 2), only 3 coincided with those for MOG20-aH2B antibodies (Table 3).

The immunization of mice with MOG (MOG20-aH3; Table 3) led to a decrease in the number of H2A hydrolysis sites with IgGs against the H3 histone (Con-aH3-0d; Table 2) from 15 to 12. It is interesting that among these sites, there were no identical ones. Somewhat unexpectedly, of the 12 sites corresponding to hydrolysis of H2A by IgGs against H3 histone after 60 days of spontaneous EAE development (Spont-aH2B-60d; Table 2), 6 sites coincided with those for MOG20-aH3 (Table 3).

With the spontaneous development of EAE, the number of H2A hydrolysis sites decreased from 8 to 7 (Table 2), while after the immunization of mice with MOG (MOG20-aH4), it increased to 12 (Table 3). Of the 12 sites for MOG20-aH4 (Table 3), only 1 and 3 sites coincided with those for Con-aH4-0d and Spont-aH4-60d, respectively (Table 3). These data indicate that the immunization of mice with MOG significantly affected the production of lymphocytes producing antibodies against five individual H1–H4 histones capable of hydrolyzing the H2A histone compared with the spontaneous development of EAE

IgG antibodies against MBP also hydrolyzed the H2A histone well. The sites of H2A histone hydrolysis by various IgG preparations against MBP are given in Table 3. Antibodies of 3-month-old mice against MBP (Con-aMBP) hydrolyzed the H2A histone at 19 sites (Table 3). Twenty days after immunization of mice with MOG (MOG20-aMBP), the number of hydrolysis sites was nearly the same: 21. Ten sites of H2A hydrolysis by these two preparations coincided and differed only in hydrolysis efficiency (Table 3).

Interestingly, the immunization of mice with MOG and the DNA–histone complex did not lead to a decisive change in the number of hydrolysis sites by anti-MBP IgGs compared to time zero (Table 3). Twenty days after mice immunization with the DNA–histone complex (DNA20-aMBP), the number of hydrolysis sites increased compared to zero time (Con-aMBP) from 19 to 22 (Table 3). A total of 10 of these 22 sites coincided with the same sites for IgGs of zero time (Con-aMBP) and 13 with those for MOG20-aMBP (Table 3). Sixty days after mice immunization with the DNA–histone complex, the number of H2A hydrolysis sites with anti-MBP antibodies decreased from 19 to 18 (Table 3). Ten sites of hydrolysis of H2A by IgGs at time zero (Con-aMBP) coincided with 60 days after mice immunization by the DNA–histone complex (DNA60-aMBP). In addition, several identical sites of H2A hydrolysis by the DNA60-aMBP preparation were found corresponding to those 20 days after immunization with the DNA–histone complex (10 sites, DNA20-aMBP) and antibodies corresponding to 20 days after mice treatment with MOG (9 sites, MOG20-aMBP) (Table 3). In the case of four anti-MBP antibody preparations, some of the same sites were common for the four IgG preparations, but there are individual sites for each of four preparations (Table 3). In addition, the sites of hydrolysis of H2A by antibodies against five histones after the immunization of mice with MOG differed from those for the four preparations against MBP (Table 3).

Overall, unlike IgGs against the five individual histones capable of hydrolyzing H2A and corresponding to the spontaneous development of EAE, the change in the number of H2A hydrolysis sites was significantly greater compared to the IgGs against MBP after the immunization of mice with MOG and DNA–histones. In addition, for all four IgG antibodies against MBP, a greater number of coincidence sites were observed than for IgGs against the five individual histones.

Data on changes in the sites of H2A hydrolysis by antibodies corresponding to 20 and 60 days after immunization of mice with the DNA–histone complex are shown in Table 4.

At zero time, IgGs against H1 hydrolyzed H2A histone at 7 sites and 60 days of spontaneous development of EAE (Spont-aH1-60d) at 28 sites (Table 2), while at 20 (DNA20-aH1) and 60 (DNA60-aH1) days after mice immunization with DNA–histone complex at 11 sites (Table 4). Thus, the spontaneous development of EAE led to an expansion of the repertoire of lymphocytes producing IgGs against H1 histone hydrolyzing H2A histone, while the immunization of mice with the DNA–histone complex resulted in a slight increase in the number of hydrolysis sites (Table 2 and Table 4). At the same time, only five sites in the case of Con-aH1-0d (Table 2) coincided with those for DNA20-aH1 and DNA60-aH1 (Table 4). However, only seven sites of H2A hydrolysis by DNA20-aH1 and DNA60-aH1 preparations were common, and five were different (Table 4). Interestingly, 10 and 9 out of 28 hydrolysis sites of H2A by Spont-aH-60d (Table 2) antibodies coincided with those for DNA20-aH1 and DNA60-aH1, respectively (Table 4). Surprisingly, however, the maximum expansion of the number of H2A hydrolysis sites by anti-H1 antibodies occurred during the spontaneous development of EAE.

At time zero, IgGs against H2A (Cont-aH2A-0d) hydrolyze the H2A histone at only four sites (Table 2). In contrast to antibodies against H1 histone, the treatment of mice with a DNA–histone complex led to a very strong increase in the number of sites of H2A hydrolysis by antibodies against H2A up to 27 and 22 sites, respectively, in the case of 20 (DNA20-aH2A) and 60 days (DNA60-aH2A) after mice immunization with the DNA–histone complex (Table 4). Only one and two of the four sites for Cont-aH2A-0d (Table 2) were found among 27 and 22 sites corresponding to DNA20-aH2A and DNA60-aH2A, respectively (Table 4). Interestingly, the development of EAE after mice immunization with the DNA–histone complex from 20 to 60 days led to a significant change in the type of H2A hydrolysis sites: only nine sites were common for DNA20-aH2A and DNA60-aH2A preparations (Table 4). In addition, the spontaneous development of EAE led to the appearance of IgGs that hydrolyzed H2A at a lower number of sites (17, Spont-aH2A-60d; Table 2) compared to Abs after mice treatment with the DNA–histone complex: 27 and 22 sites. The spontaneous development of EAE over 60 days led to stimulation of the production of lymphocytes producing IgGs hydrolyzing the H2A histone mainly at other sites compared to after mice treatment with DNA–histone complex: of the 17 sites for Spont-aH2A-60d (Table 2), only 6 sites coincided with those for the DNA20-aH2A and DNA60-aH2A preparations (Table 4). In addition, the situation with the production of IgGs against H2A in the hydrolysis of the H2A histone during spontaneous and induced EAE by the DNA–histone complex significantly differred from that for Abs against H1 hydrolyzing the H2A histone (Table 2 and Table 4).

During the spontaneous development of EAE from time zero (Con-aH2B-0d) to 60 days, the number of sites of H2A hydrolysis by IgGs against H2B histone (Spont-aH2B-60d) increased from 8 to 27 (Table 2). Interestingly, the treatment of mice with the DNA–histone complex led to the increase in the sites at 20 (DNA20-aH2B) and 60 (DNA60-aH2B) days after the immunization; the number of H2A hydrolysis sites with anti-H2B Abs increased to 19 and 35, respectively (Table 4). Only 6 and 4 hydrolysis sites of H2A were found for Con-aH2B-0d (Table 2) among, respectively, 19 and 35 sites in the case of DNA20-aH2B and DNA60-aH2B (Table 4). With an increase from 19 hydrolysis sites for DNA20-aH2B to 35 for DNA60-aH2B, 12 hydrolysis sites remained unchanged. Overall, during spontaneous and induced EAE by the DNA–histone complex, there were very strong changes in the number and type of sites for H2A hydrolysis by IgGs against the H2B histone. Among the 27 sites corresponding to 60 days of spontaneous EAE development (Spont-aH2B-60d, Table 2), only 13 were the same among 35 sites corresponding to 60 days after mice immunization with the DNA–histone complex (DNA60-aH2B, Table 4). Thus, both the spontaneous development of EAE and the treatment of mice with the DNA–histone complex strongly affected the changes in the differentiation profile of stem cells, leading to the production of lymphocytes producing IgGs against H2B histone, capable of hydrolyzing H2A at different sites.

With the spontaneous development of EAE, the appearance of lymphocytes synthesizing antibodies against different histones was entirely different. If the number of antibodies against H1, H2A, and H2B increased, then in the case of IgGs against H3 histone, it decreased from 15 to 12 (Table 2). Interestingly, compared to zero time (Con-aH3-0d; Table 2), which corresponded to 15 sites of H2A hydrolysis by IgGs against H3 histone at 20 days after treatment of mice with a DNA–histone complex (DNA20-aH3), it decreased to 11 sites (Table 4). However, the effect of mice immunization with the DNA–histone complex turned out to be multistage, and by 60 days after the immunization, the number of hydrolysis sites increased to 23 (Table 4). Nevertheless, even for 20 days after mice immunization, there were noticeable changes in hydrolysis sites; only three sites were found for Con-aH3-0d among the 11 sites for DNA20-aH3 (Table 4). Only four sites of hydrolysis of H2A by Con-aH3-0d and DNA60-aH3 antibodies were the same (Table 4). In the case of IgGs against H3 hydrolyzing the H2A histone corresponding to 20 and 60 days after immunization with DNA–histone complex, only 5 coincided (Table 4). The number of H2A histone hydrolysis sites by anti-H3 antibodies after 60 days of spontaneous EAE development was only 12 (Table 2). Remarkably, none of the 23 sites of H2A hydrolysis by IgGs against H3 histone (DNA60-aH3; Table 4) coincided with those for antibodies corresponding to 60 days of spontaneous development of EAE (Spont-aH2-60d; Table 1). Thus, in the case of antibodies against H3B hydrolyzing the H2A histone, different patterns of changes were observed during spontaneous and induced development of EAE by the DNA–histone complex compared to IgGs against three other histones (H1, H2A, and H2B) described above.

The number of sites of hydrolysis of H2A by antibodies against H4 histone practically did not change over 60 days of spontaneous development compared with zero time: seven (Spont-aH4-60d) and eight (Con-aH4-0d) sites, respectively (Table 2). Interestingly, the number of sites of H2A hydrolysis by antibodies against the H4 histone after immunization with the DNA–histone complex increased to 23 and 24 sites after 20 (DNA20-aH4) and 60 (DNA60-aH4) days (Table 4). The large number of sites (23 and 24) of hydrolysis of H2A by IgGs against H4 histone at 20 (DNA20-aH4) and 60 (DNA60-aH4) days after immunization coincided, respectively, with four and two sites detected for antibodies at the zero time of the experiment (Con-aH4-0d, Table 2). In the period from 20 to 60 days after immunization of mice with the DNA–histone complex, some changes still occurred in the repertoire of lymphocytes producing anti-H4 antibodies hydrolyzing the H2A histone since only 16 identical hydrolysis sites for DNA20-aH4 and DNA60-aH4 were found. In addition, the hydrolysis of H2A by these IgG preparations differred in the efficiency of the site cleavage (Table 4). 

## 3. Discussion

The polyreactivity or polyspecificity of complex formation characterizing different Abs is widespread [61,62,63,64]. The affinity of Abs for non-specific foreign antigens is usually significantly lower than for particular cognate ones, and such antibodies can traditionally be eluted from affinity sorbents by 0.1–0.15 M NaCl [19,20,21,22,23]. Therefore, we eluted non-specifically weakly bound Abs from affinity sorbents using 0.2 M NaCl. IgG preparations against MBP and five histones were additionally passed through alternative affinity sorbents. Finally, IgG fractions were obtained against five histones and MBP containing no alternative IgGs. Then, the fraction containing Abs against five histones was used to isolate IgGs against each of the five individual histones.

It has been previously shown that polyclonal IgG preparations from EAE-prone mice used in this study do not contain any classical proteases [46]. As previously shown, the pools of monoclonal antibodies of patients with AIDs have abzymes with serine-, thiol-like, and metal-dependent protease activities, which, unlike canonical proteases, hydrolyze proteins only at specific sites of protein sequences for which they have increased affinity [66,67,68,69,70,71]. The comparison of H2A cleavage sites with IgGs–abzymes against MBP and five individual histones well sustains this conclusion (Figure 7, Figure 8, Figure 9, Figure 10, Figure 11 and Figure 12, Table 2, Table 3 and Table 4). Trypsin hydrolyzes various proteins after arginine (R) and lysine (K) residues. The H2A sequence contains 13 sites for potential cleavage of H2A by trypsin after K and 13 sites after R. Abzymes do not hydrolyze H2A at all, but at a limited number of sites after K and R located in specific clusters of the protein sequence of this histone. In addition, the particular sites of hydrolysis after K and R for all preparations of antibodies against the five individual histones and MBP are very different. This indicates that the pools of antibodies against five histones (H1, H2A, H2B, H3, and H4) contain small fractions of histone sequence-specific serine-like abzymes.

Chymotrypsin hydrolyzes proteins after F, Y, and W aromatic AAs. The H2A protein sequence contains 1 and 3 AA residues of F and Y, respectively. None of the IgG preparations cleaved H2A after F, and only some of them hydrolyzed this histone after Y (Figure 7, Figure 8, Figure 9, Figure 10, Figure 11 and Figure 12, Table 2, Table 3 and Table 4). It can be assumed that only in some cases can the formation of sequence-specific abzymes with active centers similar to chymotrypsin occur.

The obtained data on hydrolysis sites indicate that, in addition to IgGs with serine-like active centers that mimic those of trypsin and chymotrypsin, abzymes with other alternative active centers are formed. In contrast to classical proteases, hydrolysis sites are substantially grouped in specific clusters of H2A histone sequence. They mainly occurred after neutral AAs: Q, T, G, A, L, and V (Figure 7, Figure 8, Figure 9, Figure 10, Figure 11 and Figure 12, Table 2, Table 3 and Table 4). The specific sites of H2A cleavage by all IgG preparations against five histones and MBP were not disposed of along the entire H2A histone length. Still, they were grouped into particular amino acid clusters. Depending on the IgG preparation, they were located not only in the same or overlapping sequences, but in sequences located in different zones of the histone H2A protein sequence (Figure 7, Figure 8, Figure 9, Figure 10, Figure 11 and Figure 12, Table 2, Table 3 and Table 4).

The decisive proof that the abzymes against MBP and five individual histones did not contain even perceptible impurities of Abs against alternative histones arises from the very different number of specific sites of H2A hydrolysis (from 4 to 35) by IgGs–abzymes against each of the five individual histones and MBP (Figure 7, Figure 8, Figure 9, Figure 10, Figure 11 and Figure 12, Table 2, Table 3 and Table 4). Interestingly, the same major hydrolysis sites were found for some abzyme preparations, along with individual ones, but there was no common hydrolysis site for all 29 IgG preparations (Table 2, Table 3 and Table 4). All splitting sites differred not only in their location in the H2A sequence but also in the same hydrolysis sites found in the case of different IgGs that varied greatly in the relative efficiency of hydrolysis: major, medium, or minor sites (Figure 7, Figure 8, Figure 9, Figure 10, Figure 11 and Figure 12, Table 2, Table 3 and Table 4). 

All IgG preparations differred in the number and type of cleavage sites and the hydrolysis efficiency of the same coinciding sites (Figure 7, Figure 8, Figure 9, Figure 10, Figure 11 and Figure 12, Table 2, Table 3 and Table 4). The analysis of which hydrolysis sites were most often considered major in the case of all preparations used is of particular interest. For 19 out of 29 IgG preparations, there was one identical major hydrolysis site: the R17-S18 site. Other major sites were less common among H2A hydrolysis sites in the case of different IgG reparations (number of preparations): I87-R88 (8), Q104-G105 (7), V114-L115 (7), Q84-L85 (5), E121-S122 (4), Y50-L51 (4), E121-S122 (4), E64-L65 (3), R77-I78 (3), R81-H82 (3), R35-K36 (2), S40-E41 (2), P48-V49 (2), L51-A52 (2), Y57-L58 (2), G67-N68 (2), N73-K74 (2), and L85-A86 (2). Interestingly, original major hydrolysis sites specific only to one IgG preparation, which were absent in the case of other antibodies, were found in the case of 19 antibody preparations. Therefore, it is interesting to understand why at different stages of spontaneous EAE development and after the immunization of mice with MOG and the DNA–histone complex the formation of IgGs hydrolyzing the H2A histone at many different sites is possible. For this, it is helpful to take into account some literature data.

First, MS of people is at least a two-phase autoimmune disease [72]. The cascade of many reactions at the first inflammatory phase is very sophisticated, involving many cytokines, chemokines, enzymes, proteins, and other compounds inducing macrophages and other cells producing NO^●^ radicals and osteopathin [72]. The coordinated constituent action of B- and T-cells, complementary systems, mediators of inflammation, and auto-antibodies leads to the formation of demyelination nidi and infringement of axon conductivity. The neurodegenerative stage of MS appearing later is directly associated with the patient’s neural tissue destruction [72]. Therefore, during the analysis of immunological, biochemical, and clinical indices of MS, every current phase of the disease must be considered, including changes in immunoregulation, the exhaustion of different compensatory and adaptive mechanisms, and systemic metabolic changes [72]. It should be assumed that the spontaneous and antigen-induced development of EAE in mice can also proceed in several stages, when at different stages, auto-antigens, including histones, and MBP, can form complexes with a variety of proteins, nucleic acids, lipids, polysaccharides, cells, etc.

When analyzing the diversity in the number and type of sites of H2A hydrolysis by Abs against five individual histones and MBP, it is important to take into account the spontaneous development of autoimmune reactions associated with a change in the differentiation profile of bone marrow stem cells occurring in several stages. The beginning and acute phases are the first and second stages, and with the transition to a deep pathology, the next stage of changes in this differentiation profile can be observed [10,11,12,13,14,15,16]. The treatment of mice with MOG and the DNA–histone complex strongly alters the pattern of HSC differentiation compared to its transformation during the spontaneous development of EAE [10,11,12,13,14,15,16]. 

It is known that the formation of natural abzymes occurs in the case of the structures of various molecules that imitate, at least to some extent, the transition states of chemical reactions [17,18,19,20,21,22,23]. In principle, abzymes can be formed in the case of any of the antigenic determinants or sequences of a protein that mimic the transition state of a peptide bond hydrolysis reaction. In addition, a variety of compounds can affect the immunogenicity of free histones, improving or impairing their ability to stimulate antibody production.

Importantly, all histones have several antigenic determinants [73,74,75,76]. The H2A histone has at least 11 antigenic regions [77], while MBP has 4 antigenic determinants [77,78]. At the same time, the antigenic determinants of all histones have a high level of homology between themselves and with those of MBP [44,45,46]. Like histones, MBP binds efficiently to DNA [79]. Thus, this may be the main reason why protein sequences of histones and MBP with altered structures in various types of complexes can mimic the transition states of hydrolytic reactions and stimulate the formation of abzymes with catalytic cross-reactivity.

At all stages of EAE development, the apoptosis of various cells constantly occurs, and chromatin DNA–histone complexes enter the blood. This leads to the production of Abs against DNA itself or individual histones. However, there may be some quite different variants of such complexes. DNA in complex with histones can sterically shield some epitopes and, at the same time, increase the immunogenicity of other sequences of histones available for stimulation of abzymes formation. In addition, the five histones within such complexes can form antigenic determinants formed at the junction of two or even three histones. In this case, recognition sites of abzymes may be chimerically capable of binding and hydrolyzing several histones.

Moreover, it has recently been shown that antigenic determinants are formed at the junction of protein sequences of histones and DNA, the formation of antibodies against which leads to the production of abzymes that hydrolyze both DNA and histones [72]. Individual histones, their associates, and complexes with DNA can interact with a variety of compounds characteristic of each of the different stages and sub-stages of EAE development. This can lead to the formation of a massive number of very different abzymes against individual histones and their complexes with each other and with other blood components. The efficient hydrolysis of the histone H4 by EAE-prone mice IgGs against the five histones H1–H4 has recently been shown [46]. This work proved that antibodies against all five histones and MBP effectively hydrolyze the H2A histone. In addition to the above pathways for the formation of antibodies demonstrating catalytic cross-reactivity against various histones and MBP, another reason for this phenomenon may be the high level of protein sequence homology of all five histones and MBP [44,45]. Thus, the formation of polyreactivity of complexation and catalytic cross-reactivity of abzymes against histones and MBP can occur in several parallel ways.

## 4. Materials and Methods

### 4.1. Materials and Chemicals

All compounds used, H2A (M2502S), H1 (M2501S), H2B (M2508S), H4 (M2504S), H3 (M2503S), and an equimolar mixture of the histones (H9250) were from Sigma (St. Louis, MO, USA). Protein G–Sepharose (17061801) and Superdex 200 HR 10/30 columns (17061801) were from GE Healthcare (New York, NY, USA). MBP was obtained from the Center of Molecular Diagnostics and Therapy (DBRC-HMBP; Moscow, Russia). Affinity sorbents containing immobilized MOG, mixture, and five individual histones were obtained according to the standard manufacturer’s protocol using BrCN-activated Sepharose from GE Healthcare (17098101; New York, NY, USA), the mixture of five histones or 5 individual histones and MBP. Mouse oligopeptide MOG_35–55_ was bought from EZBiolab (Heidelberg, Germany). All preparations used were free from any possible contaminants

### 4.2. Experimental Animals

Three-month-old inbred C57BL/6 mice were recently used by us to analyze possible mechanisms of spontaneous and antigen-induced development of EAE [11,12,46]. They were raised using standard conditions free of pathogens in a special mouse vivarium of the Institute of Cytology and Genetics (ICG). All experiments with C57BL/6 mice were implemented under the Bioethical Committee of the ICG protocols (number of the document—134A of 7 September 2010), satisfying the humane principles for experiments with animals of European Communities Council Directive: 86/609/CEE. This Committee of the Institute supported this study. The relative weights, titers of Abs against histones and MBP, the concentration of the urine proteins, mg/mL (proteinuria), and some other indexes characterizing progress in EAE development were analyzed and described in [11,12,13,14,46]. 

### 4.3. Antibody Purification

Electrophoretically homogeneous preparations of polyclonal IgGs from the blood plasma of C57BL/6 mice were first isolated by plasma protein affinity chromatography on Protein G–Sepharose. The IgG preparations were additionally purified by FPLC (fast protein liquid chromatography–gel filtration) on Superdex-200 HR 10/30 column [11,12,13,14,46]. After gel filtration for additional purification of IgGs, central parts of Abs peaks were filtered through filters (pore size 0.1 µm) as in [11,12,13,14,46]. 

Removal of all antibodies against five histones (H1–H4) from total IgG preparations was fulfilled using histone5His–Sepharose (5 mL) containing immobilized five H1–H4 histones. The column was equilibrated with buffer A (20 mM Tris–HCl, pH 7.5). After antibody loading, the column was washed with A buffer to zero optical density (A_280_). Non-specifically adsorbed antibodies possessing a low affinity for five histones were eluted using A buffer supplemented with NaCl (0.2 M). Then, IgGs with high affinity for histones were specifically desorbed using an acidic buffer (0.1 M glycine-HCl, pH 2.6). The IgGs eluted from histone5His–Sepharose at loading and washing of the column with buffer A (5 mL) were unified and used to isolate IgGs against MBP by their chromatography on the 5 mL MBP–Sepharose column equilibrated with buffer A. After the MBP–Sepharose washing with A buffer to zero optical density (A_280_), adsorbed low affinity for MBP IgGs were eluted first using buffer A and then with the buffer containing NaCl (0.2 M). Then, anti-MBP IgGs were eluted from the sorbent by acidic Tris–Gly buffer (pH = 2.6), similar to histone5His–Sepharose. Further, this fraction of antibodies was named and used as anti-MBP antibodies. Such IgGs were obtained in the case of mice corresponding to different stages of development of EAE before (Con-aMBP-0d, Spont-aMBP-60d), after immunization of mice with MOG (MOG20-aMBP) and DNA–histone complex (DNA20-aMBP and DNA60-aMBP) (Tab1e 1). 

For additional purifications of IgGs against five histones from possible hypothetical small impurities of Abs against MBP, the fractions eluted from histone5His–Sepharose were re-chromatographed on MBP–Sepharose. IgGs eluted at the loading on MBP–Sepharose were named anti-5-histone IgGs. IgGs against 5 histones were obtained from plasma of the blood of mice corresponding to different stages of development of EAE before (Con-aH1-H4-0d, Spont-aH1-H4-60d), after immunization of mice with MOG (MOG20-aH1-H4-20) and DNA–histone complex (DNA20-aH1-H4 and DNA60-aH1-H4) (Table 1). 

### 4.4. Antibody Purification against Individual Five Histones

All five preparations against five histones, corresponding to different stages of EAE development, were further used to obtain IgGs against each of the five individual histones. All of them were applied sequentially, first on H2A–Sepharose containing immobilized H2A histone. The fractions eluted at loading were then sequentially applied to the following four sorbents: H1–Sepharose, H2B–Sepharose, H3–Sepharose, and H4–Sepharose. All affinity chromatographies were performed as in the case of histone5His–Sepharose and MBP–Sepharose. IgGs against H2A and other histones were specifically desorbed from each of five affinity sorbents with buffer, pH 2.6. These IgG fractions were designated, respectively, as anti-H2A, anti-H1, anti-H2B, anti-H3, and anti-H4 IgGs with the denotation to which stage and which antigen they correspond: Con (beginning of the experiment); Spont (spontaneous of EAE development); MOG (mice treated with MOG); DNA (mice treated with DNA–histone complex) (Table 1).

### 4.5. Proteolytic Activity Assay

Protease activity of all IgGs–abzymes was analyzed by SDS-PAGE using the reaction mixtures (10–18 μL) containing Tris–HCl buffer (20 mM; pH 7.5), 0.85–1.0 mg/mL MBP, the mixture of five histones, or H2A histone, and 0.07-0.1 mg/mL IgGs against MBP or 0.012 mg/mL Abs against five histones as described in [46]. All mixtures of histones and IgGs were incubated over 3–14 h at 37 °C. Then, all reactions were stopped by adding SDS to the 0.1% final concentration. The efficiency of H2A and MBP hydrolysis was analyzed by SDS-PAGE in 20% gel. All gels were colored using silver or Coomassie Blue. The relative protease activities of IgGs were evaluated from the decrease in relative intensity of protein bands corresponding to initial non-hydrolyzed H2A or MBP corresponding to these proteins incubation without Abs. A more detailed analysis of the hydrolysis of H2A by all obtained Abs was carried out using the MALDI-TOF spectrometry method.

### 4.6. MALDI-TOF Analysis of Histones Hydrolysis

H2A histone was hydrolyzed over 0–20 h using all preparations of anti-MBP and against 5 individual histones IgGs using conditions described above. The H2A histone hydrolysis products were ascertained by the 337-nm nitrogen laser VSL-337 ND, 3 ns pulse duration of Reflex III system (Bruker, Frankfurt, Germany). Small aliquots of reaction mixtures (1–2 µL) were used for analysis after reaction mixtures incubation at different times by MALDI mass spectrometry. Sinapinic acid was used as the matrix. To 1.7 µL of the matrixes and 1.7 µL of 0.2% trifluoroacetic acid, 1.7 µL of the solutions containing histone H2A were added, and 1.0–1.7 µL of the obtained mixtures were applied to the iron MALDI plates. For the analysis, they were air-dried. All MALDI spectra were calibrated using standards II and I calibrant mixtures of special oligopeptides and proteins (Frankfurt, Germany, Bruker Daltonic) in the external and/or internal calibration mode. The analysis of molecular weights and specific sites of H2A hydrolysis by various IgGs was done using Protein Calculator v3.3 (Scripps Research Institute). 

### 4.7. Statistical Analysis

The results correspond to the average values (mean ± standard deviation) from 8–10 independent spectra for each preparation of IgGs against five individual histones and MBP. 

## 5. Conclusions

Here, we first demonstrated that IgGs–abzymes from EAE-prone C57BL/6 mice against five individual histones and myelin basic protein can form complexes with H2A histone showing polyreactivity in complexation. In addition, all IgG preparations possessed unusual enzymatic cross-reactivity in the hydrolysis of the histone H2A. The IgGs against five individual histones and MBP at the beginning of the experiment (3-old-mice) and after spontaneous development of EAE over 60 days differred in the relative number of hydrolysis sites and their type. After mice immunization with MOG or the DNA–histone complex, the sites of H2A hydrolysis differred from each other, as well as from those for Abs corresponding to the beginning of the experiment and the spontaneous development of EAE. Overall, the number of H2A hydrolysis sites depending on the IgG preparation varied from 4 to 35. The treatment of mice with MOG and the DNA–histone complex accelerated EAE evolution and led to a change in the relative activity of IgGs in H2A hydrolysis. The possible reasons for the catalytic cross-reactivity of antibodies and such strong differences in the number and type of histone H2A cleavage sites were analyzed.

## Figures and Tables

**Figure 1 ijms-24-08636-f001:**
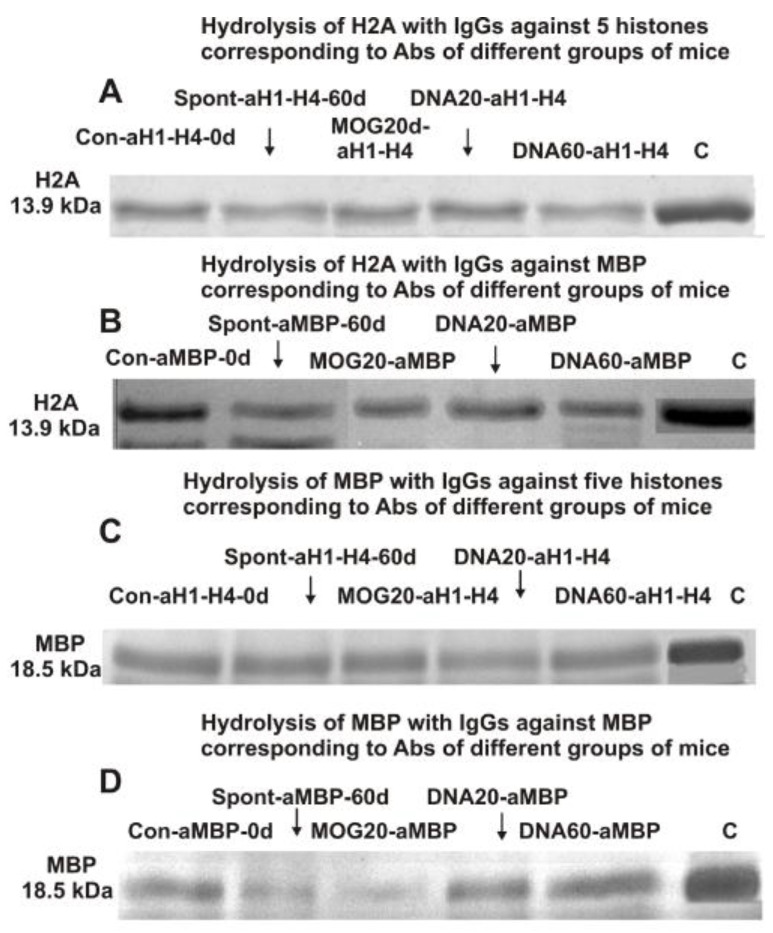
SDS-PAGE analysis of H2A histone hydrolysis by IgGs–abzymes against five histones (**A**) and this histone with IgGs against MBP (**B**) as well as splitting myelin basic protein by IgGs against five histones (**C**) and IgGs–abzymes against MBP (**D**). Lane C corresponds to the histones (**A**,**B**) and MBP (**C**,**D**) incubated without IgGs. MBP and a mixture of five histones with and without IgGs (0.03 mg/mL) were incubated for 14 h.

**Figure 2 ijms-24-08636-f002:**
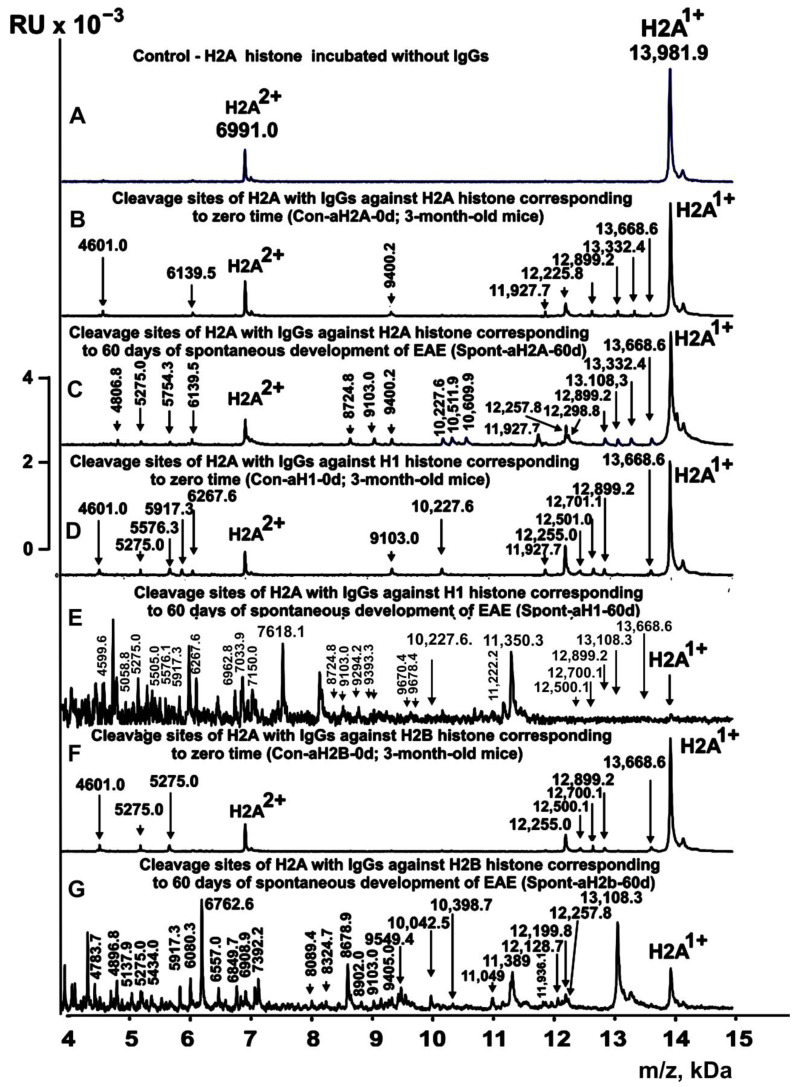
MALDI spectra corresponding to products of H2A histone (0.85 mg/mL) hydrolysis in the absence (**A**) and the presence of IgGs (0.04 mg/mL) against three histones: Con-aH2A-0d (**B**), Spont-aH2A-60d (**C**), Con-aH1-0d (**D**), Spont-aH1-60d (**E**), Con-aH2B-0d (**F**), and Spont-aHB-60d (**G**). All designations of IgG preparations and the values of *m*/*z* are shown in the Figure.

**Figure 3 ijms-24-08636-f003:**
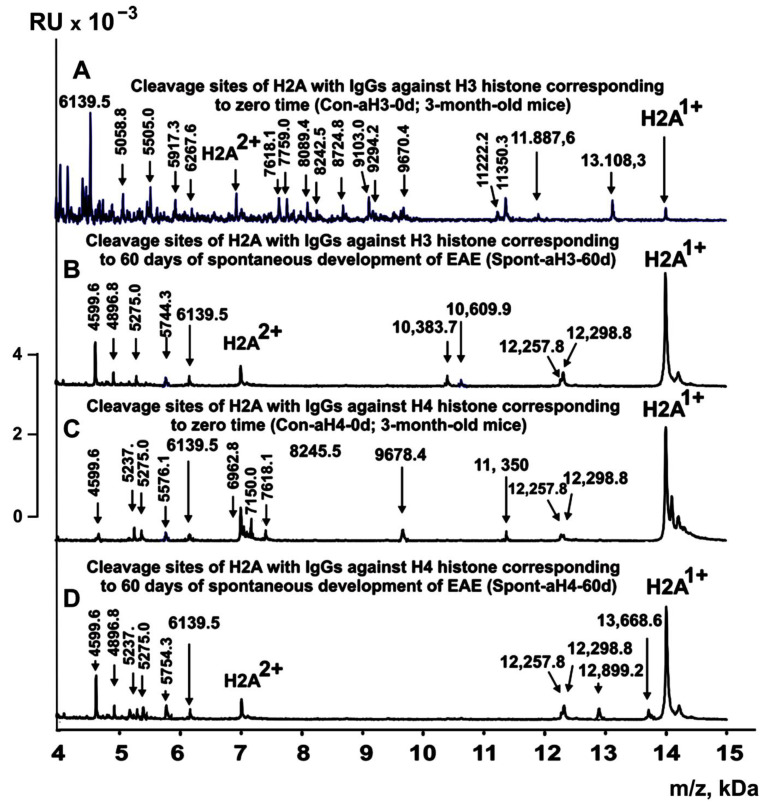
MALDI spectra corresponding to products of H2A histone (0.85 mg/mL) hydrolysis in the presence of IgGs (0.04 mg/mL) against two (H3 and H4) histones: Con-aH3-0d (**A**), Spont-aH3-60d (**B**), Con-aH4-0d (**C**), and Spont-aH4-60d (**D**). All designations of IgG preparations and the values of *m*/*z* are shown in the Figure.

**Figure 4 ijms-24-08636-f004:**
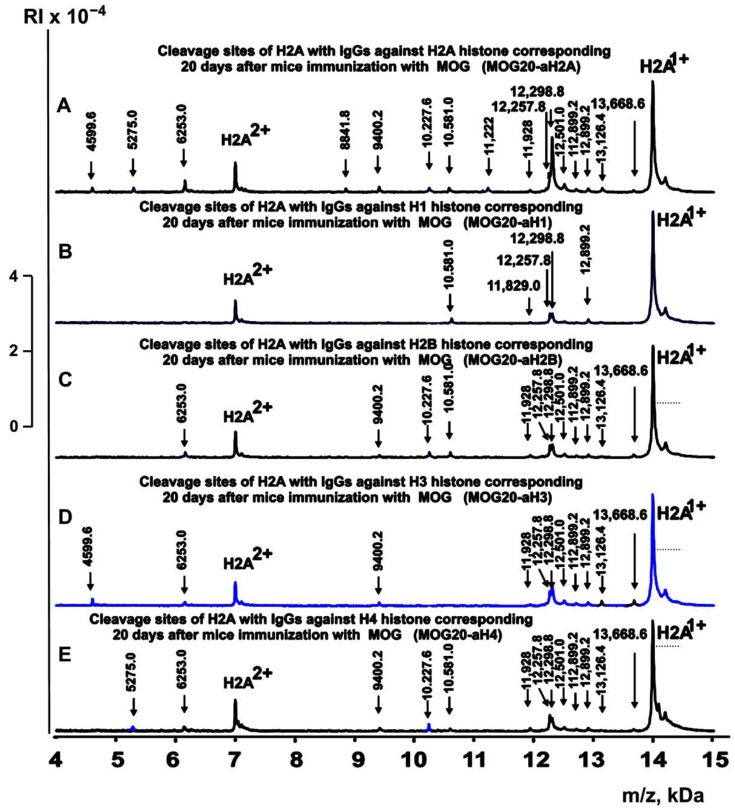
MALDI spectra of H2A (0.85 mg/mL) hydrolysis products with IgGs (0.04 mg/mL) against five histones corresponding to 20 days after mice immunization with MOG: MOG20-aH2A (**A**), MOG20-aH1 (**B**), MOG20-aH2B (**C**), MOG20-aH3 (**D**), and MOG20-aH4 (**E**). All designations of IgG preparations and the values of *m*/*z* are shown in the Figure.

**Figure 5 ijms-24-08636-f005:**
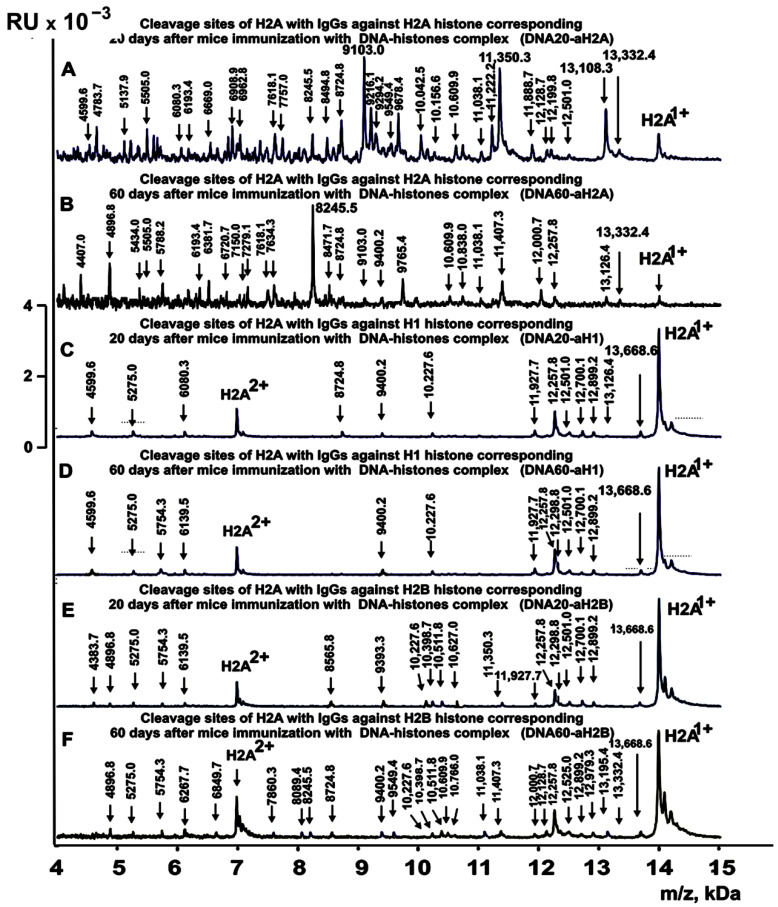
MALDI spectra of H2A (0.85 mg/mL) hydrolysis products with IgGs (0.04 mg/mL) against three histones corresponding to 20 and 60 days after mice immunization with DNA–histone complex: DNA20-aH2A (**A**), DNA60-aH2A (**B**), DNA20-aH1 (**C**), DNA60-aH1 (**D**), DNA20-aH2B (**E**), and DNA60-aH2B (**F**). All designations of IgG preparations and the values of *m*/*z* are shown in the Figure.

**Figure 6 ijms-24-08636-f006:**
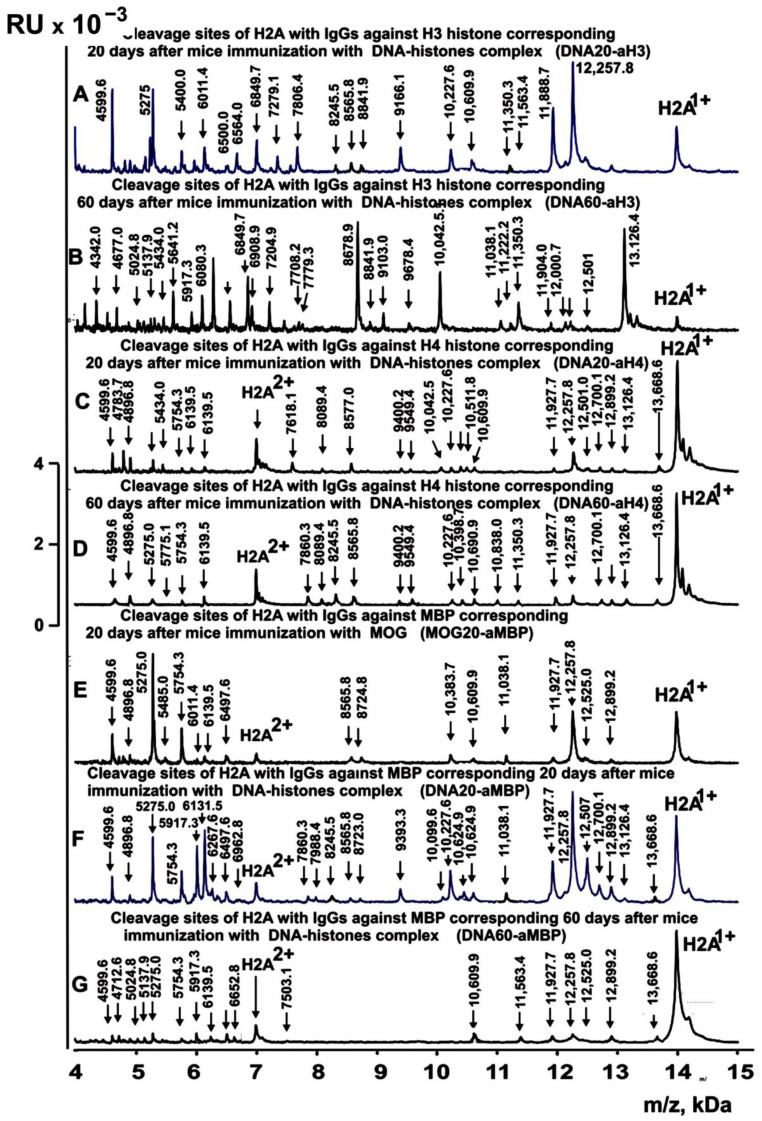
MALDI spectra of H2A (0.85 mg/mL) hydrolysis products with IgGs (0.04 mg/mL) against two H3 and H4 histones and MBP corresponding to 20 and 60 days after mice immunization with DNA–histone complex: DNA20-aH3 (**A**), DNA60-aH3 (**B**), DNA20-aH4 (**C**), DNA60-aH4 (**D**), DNA20-aMBP (**E**), and DNA20-aMBP (**F**) and DNA60-aMBP (**G**). All designations of IgG preparations and the values of *m*/*z* are shown in the Figure.

**Figure 7 ijms-24-08636-f007:**
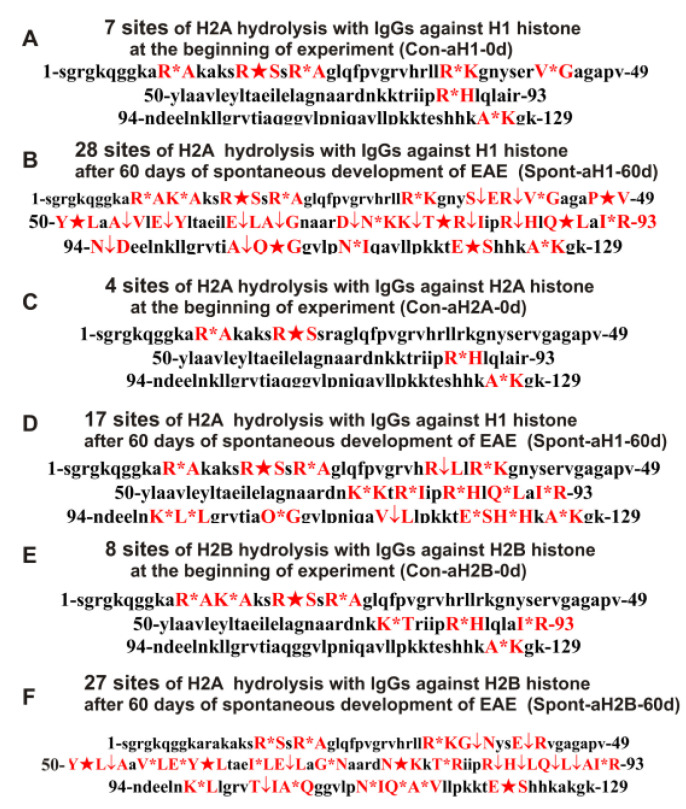
Sites of H2A hydrolysis (in red) by IgGs against three histones (H2A, H1, and H2B) corresponding to zero time (3-month-old mice) and after spontaneous development of EAE (before mice immunization) over 60 days: Con-aH1-0d (**A**), Spont-aH1-60d (**B**), Con-aH2A-0d (**C**), Spont-aH2A-60d (**D**), Con-aH2B-0d (**E**), and Spont-aH2B-60d (**F**). Major sites of H2A cleavage are shown by stars (★), moderate ones by arrows (↓), and minor sites of the cleavages by small stars (*).

**Figure 8 ijms-24-08636-f008:**
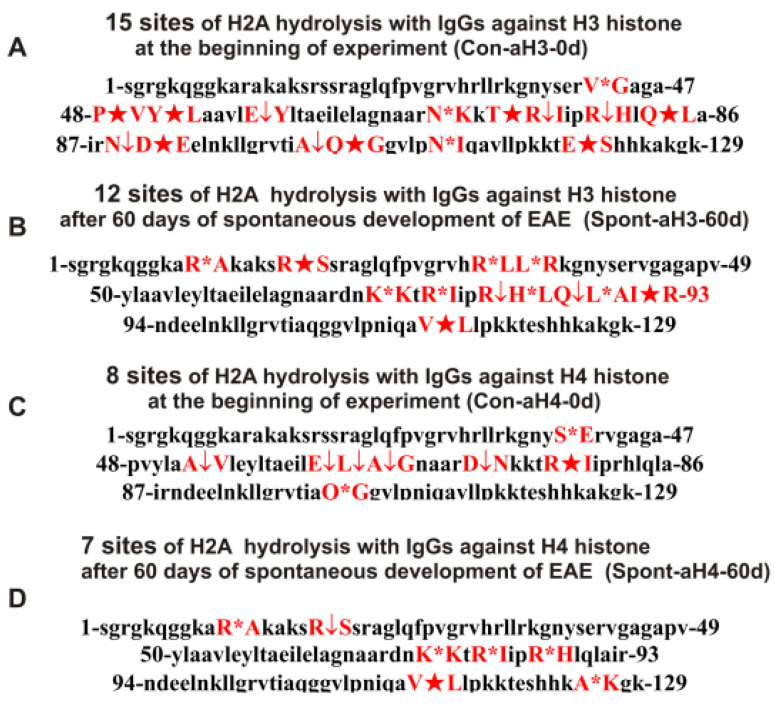
Sites of H2A hydrolysis (in red) by IgGs against H3 and H4 histones corresponding to zero time (3-month-old mice) and after spontaneous development of EAE (before mice immunization) over 60 days: Con-aH3-0d (**A**), Spont-aH3-60d (**B**), Con-aH4-0d (**C**), and Spont-aH4-60d (**D**). Major sites of H2A cleavage are shown by stars (★), moderate ones by arrows (↓), and minor sites of the cleavages by small stars (*).

**Figure 9 ijms-24-08636-f009:**
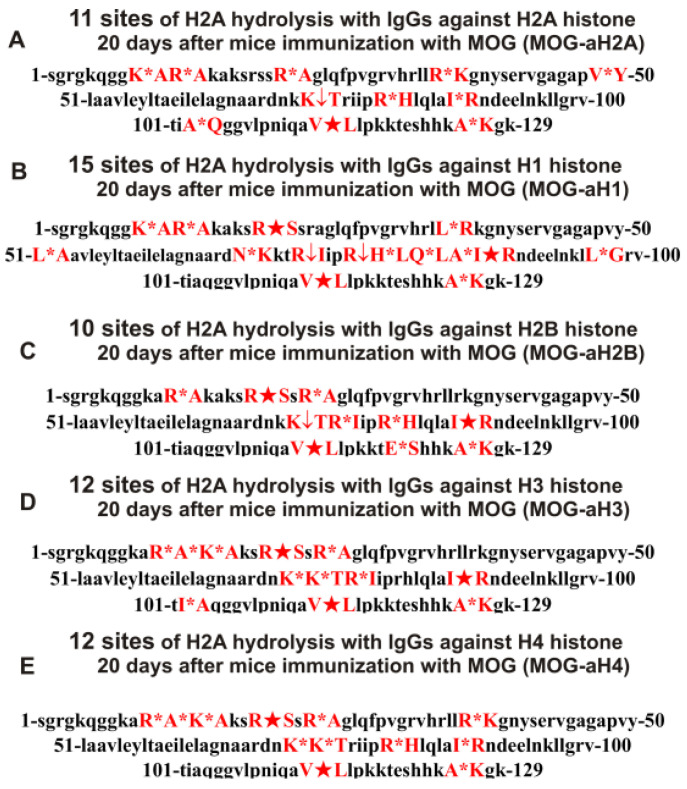
Sites of H2A hydrolysis (in red) by IgGs against five H1–H4 histones corresponding to 20 days after mice immunization with MOG: MOG20-aH2A (**A**), MOG20-aH1 (**B**), MOG20-aH2B (**C**), MOG20-aH3 (**D**), and MOG20-aH4 (**E**). Major sites of H2A cleavage are shown by stars (★), moderate ones by arrows (↓), and minor sites of the cleavages by small stars (*).

**Figure 10 ijms-24-08636-f010:**
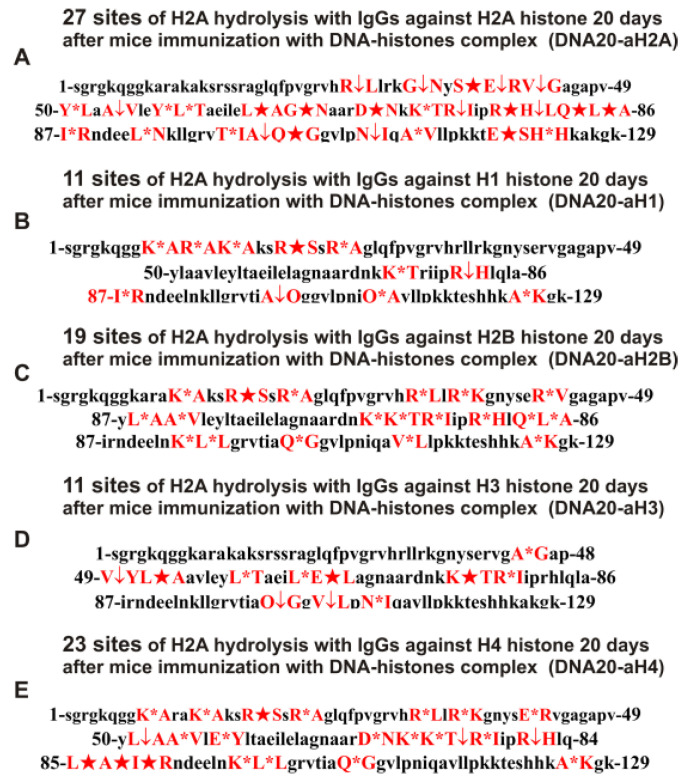
Sites of H2A hydrolysis (in red) by IgGs against five individual H1–H4 histones corresponding to 20 days after mice immunization with DNA–histone complex: DNA20-aH2A (**A**), DNA20-aH1 (**B**), DNA20-aH2B (**C**), DNA20-aH3 (**D**), and DNA20-aH4 (**E**). Major sites of H2A cleavage are shown by stars (★), moderate ones by arrows (↓), and minor sites of the cleavages by small stars (*).

**Figure 11 ijms-24-08636-f011:**
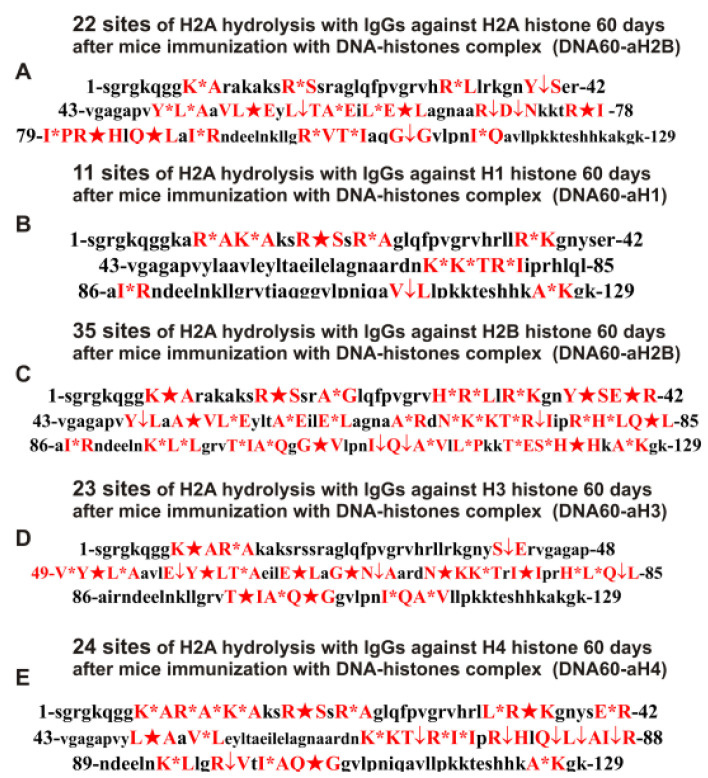
Sites of H2A hydrolysis (in red) by IgGs against five H1–H4 histones corresponding to 60 days after mice immunization with DNA–histone complex: DNA60-aH2A (**A**), DNA60-aH1 (**B**), DNA60-aH2B (**C**), DNA60-aH3 (**D**), and DNA60-aH4 (**E**). Major sites of H2A cleavage are shown by stars (★), moderate ones by arrows (↓), and minor sites of the cleavages by small stars (*).

**Figure 12 ijms-24-08636-f012:**
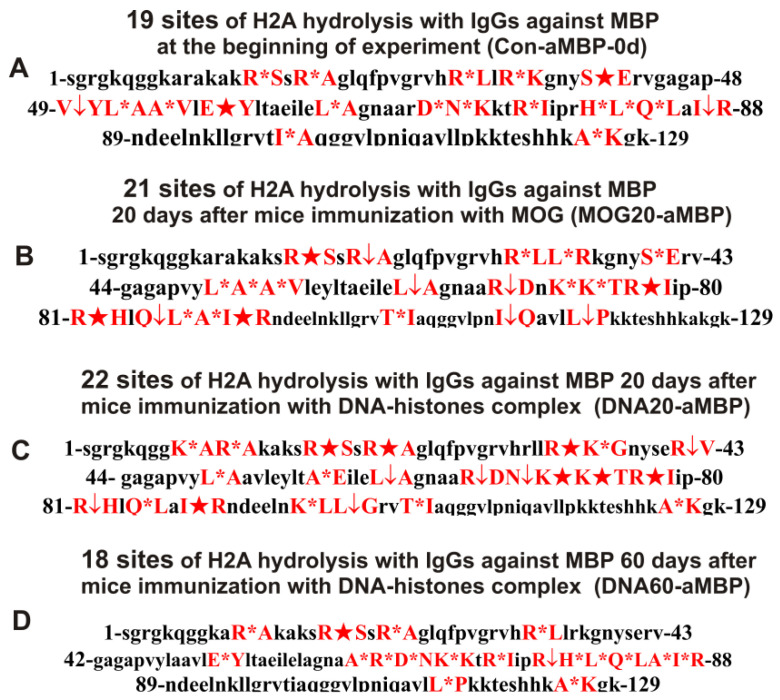
Sites of H2A hydrolysis (in red) by IgGs against MBP corresponding to zero time (3-month-old mice), 20 days after mice treatment with MOG as well as 20 and 60 days after mice immunization with DNA–histone complex: Con-aMBP-0d (**A**), MOG20-aMBP (**B**), DNA20-aMBP (**C**), and DNA60-aMBP (**D**). Major sites of H2A cleavage are shown by stars (★), moderate ones by arrows (↓), and minor sites of the cleavages by small stars (*).

**Table 1 ijms-24-08636-t001:** IgGs against 5 histones (total), MBP, and individual histones corresponding to different stages of EAE development *.

Zero time (control), beginning of experiments (3 mice-age mice)	Different total IgGs	Designation	IgGs against individual histones	Designation
	Zero time (control) beginning of the experiment, IgGs against 5 histones and MBP	Con-aH1-H4-0d **	anti-H1 histone	Con-aH1-0d
			anti-H2A histone	Con-aH2A-0d
			anti-H2B histone	Con-aH2B-0d
		Con-aMBP-0d	anti-H3 histone	Con-aH3-0d
			anti-H4 histone	Con-aH4-0d
Spontaneous development of EAE over 60 days (without mice immunization at 3 mice of age)	Spontaneous development of EAE over 60 days; IgGs against 5 histones and MBP	Spont-aH1-H4-60d	anti-H1 histone	Spont-aH1-60d
			anti-H2A histone	Spont-aH2A-60d
			anti-H2B histone	Spont-aH2B-60d
		Spont-aMBP-60d	anti-H3 histone	Spont-aH3-60d
			anti-H4 histone	Spont-aH4-60d
IgGs corresponding to 20 days after mice immunization with MOG	IgGs against 5 histones and MBP corresponding to 20 days after mice immunization with MOG	MOG20-aH1-H4-20	anti-H1 histone	MOG20-aH1
			anti-H2A histone	MOG20-aH2A
			anti-H2B histone	MOG20-aH2B
		MOG20-aMBP	anti-H3 histone	MOG20-aH3
			anti-H4 histone	MOG20-aH4
IgGs corresponding to 20 days after mice immunization with DNA–histone complex	IgGs against 5 histones and MBP; 20 days after mice immunization with DNA–histone complex	DNA20-aH1-H4	anti-H1 histone	DNA20-aH1
			anti-H2A histone	DNA20-aH2A
			anti-H2B histone	DNA20-aH2B
		DNA20-aMBP	anti-H3 histone	DNA20-aH3
			anti-H4 histone	DNA20-aH4
IgGs corresponding to 60 days after mice immunization with DNA–histone complex	IgGs against 5 histones and MBP; 60 days after mice immunization with DNA–histone complex	DNA60-aH1-H4	anti-H1 histone	DNA60-aH1
			anti-H2A histone	DNA60-aH2A
			anti-H2B histone	DNA60-aH2B
		DNA60-aMBP	anti-H3 histone	DNA60-aH3
			anti-H4 histone	DNA60-aH4

* All these IgG preparations were used to analyze H2A histone hydrolysis. ** In terms of preparations, Con corresponds to control IgG preparations: zero time of the experiment (3-month-old mice); Spont-60: spontaneous development of EAE within 60 days after the start of the experiment; MOG20: 20 days after mice treatment with MOG; DNA20 and DNA60: 20 and 60 days after mice immunization with the DNA–histone complex, respectively. In addition, the short designations of the preparations indicate against which antigen the antibodies act: aH1–aH4: anti-H1–anti-H4 histones; aMBP: anti-MBP; aH1: anti-H1; aH2A: anti-H2A; aH2B: anti-H2B; aH3: anti-H3; aH4: anti-H4.

**Table 2 ijms-24-08636-t002:** Sites of H2A histone hydrolysis by IgGs against five individual histones at zero time and spontaneous development of EAE during 60 days *.

Sites of H2A Hydrolysis at the Beginning of Experiments and after 60 Days of Spontaneous Development of EAE									
Con-aH1-0d	Con-aH1-60d	Con-aH2A-0d	Con-aH2A-60d	Con-aH2B-0d	Con-aH2B-60d	Con- aH3-0d	Con- aH3-60d	Con- aH4-0d	Con- aH4-60d
7 Sites	28 Sites	4 Sites	17 Sites	8 Sites	27 Sites	15 Sites	12 Sites	8 Sites	7 Sites
R11-A12	R11-A12 **	R11-A12	R11-A12	-	-	-	R11-A12	-	R11-A12
-	-	-	-	R11-A12	-	-	-	-	-
-	K13-A14	-	-	-	-	-	-	-	-
-	-	-	-	K14-A14	-	-	-	-	-
R17-S18 **	R17-S18	R17-S18	R17-S18	R17-S18	R17-S18	-	R17-S18	-	R17-S18
R20-A21	R20-A21	-	R20-A21	R20-A21	R20-A21	-	-	-	-
-	-	-	R32-L33 **	-	-	-	R32-L33	-	-
-	-	-	-	-	-	-	L33-R34	-	-
R35-K36	R35-K36	-	R35-K36	-	R35-K36	-	-	-	-
-	-	-	-	-	G37-N38	-	-	-	-
-	S40-E41	-	-	-	E41-R42	-	-	S40-E41	-
-	R42-V43	-	-	-	-	-	-	-	-
V43-G44	V43-G44	-	-	-	-	V43-G44	-	-	-
-	P48-V49	-	-	-	-	P48-V49	-	-	-
-	Y50-L51	-	-	-	Y50-L51	Y50-L51	-	-	-
-		-	-	-	L51-A52	-	-	-	-
-	A53-V54	-	-	-		-		A53-V54	-
-		-	-	-	V54-L55	-	-	-	-
-	E56-Y57	-	-	-	E56-Y57	E56-Y57	-	-	-
-		-	-	-	Y57-L58	-	-	-	-
-		-	-	-	I62-L63	-	-	-	-
-	E64-L65	-	-	-	E64-L65	-	-	E64-L65	-
-		-	-	-		-	-	L65-A66	-
-	A66-G67	-	-	-		-	-	A66-G67	-
-		-	-	-	G67-N68	-	-	-	-
-	D72-N73	-	-	-		-	-	D72-N73	-
-	N73-K74	-	-	-	N73-K74	N73-K74	-	-	-
-	-	-	K74-K75		-	-	K74-K75	-	K74-K75
-	K75-T76	-	-	K75-T76	-	-	-	-	-
-	T76-R77	-	-	-	T76-R77	-	-	-	-
-	R77-I78	-	-	-	-	-	R77-I78	-	R77-I78
-	R81-H82	-	R77-I78	-	R81-H82	-	R81-H82	-	R81-H82
-	-	-	-	-	H82-L83	-	H82-L83	-	-
-		-	-	-	-	-	Q84-L85	-	-
-	-	-	-	-	-	-	L85-A86	-	-
-	-	-	-	-	-	N76-R77	-	-	-
-	-	-	-	-	-	R77-I78	-	R77-I78	-
R81-H82	-	R81-H82	R81-H82	R81-H82	-	R81-H82	-	-	-
-	Q84-L85	-	Q84-L85	-	Q84-L85	Q84-L85	-	-	-
-	-	-	-	-	L85-A86	-	-	-	-
-	I87-R88	-	I87-R88	I87-R88	I87-R88	-	I87-R88	-	-
-	N89-D90	-	-	-	-	N89-D90	-	-	-
-	-	-	-	-	-	D90-E91	-	-	-
-	-	-	K95-L96	-	K95-L96	-	-	-	-
-	-	-	L96-L97	-	-	-	-	-	-
-	-	-	-	-	T101-I102	-	-	-	-
-	A103-Q104	-	-	-	A103-Q104	A103-Q104	-	-	-
-	Q104-G105	-	Q104-G105	-	-	Q104-G105	-	Q104-G105	-
-	N110-I11	-	-	-	N110-I11	N110-I11	-	-	-
-	-	-	-	-	Q112-A113	-	-	-	-
-	-	-	-	-	A113-V144	-	-	-	-
-	-	-	V114-L115	-	-	-	V114-L115	-	V114-L115
-	E121-S122	-	E121-S122	-	E121-S122	E121-S122	-	-	-
-	-	-	H123-H124	-	-	-	-	-	-
A126-K127	A126-K127	A126-K127	A126-K127	A126-K127	-	-	-	-	A126-K127

* The molecular weight of the histone hydrolysis products and the corresponding sites of the hydrolysis were determined based on a set of data from 8-10 spectra.** Major sites of the hydrolysis are shown in red, moderate sites in black, and minor sites in green. Missing splitting sites are marked with a dash (-).

**Table 3 ijms-24-08636-t003:** Sites of H2A histone hydrolysis by IgGs against five individual histones and MBP 20 dayafter mice immunization with MOG *.

Sites of H2A Hydrolysis by IgGs 20 Days after Mice Immunization with MOG and Antibodies against MBP								
MOG20-aH1	MOG20-aH2A	MOG20-aH2B	MOG20-aH3	MOG20-aH4	Con-aMBP	MOG20-aMBP	DNA20-aMBP	DNA60-aMBP
15 Sites	11 Sites	10 Sites	12 Sites	12 Sites	19 Sites	21 Sites	22 Sites	18 Sites
K9-A10 **	K9-A10	-	-	-	-	-	K9-A10	-
R11-A12	R11-A12	R11-A12	R11-A12	R11-A12	-	-	R11-A12	R11-A12
-	-	-	A12-K13	A12-K13	-	-	-	-
-	-	-	K13-A14	K13-A14	-	-	-	-
-	-	-	-	-	-	-	-	-
-	R20-A21	-	-	-	-	-	-	-
R17-S18 **	-	R17-S18	R17-S18	R17-S18	R17-S18	R17-S18	R17-S18	R17-S18
-		R20-A21	R20-A21	R20-A21	R20-A21	R20-A21	R20-A21	R20-A21
-	-	-	-	-	R32-L33	R32-L33	-	R32-L33
L34-R35	-	-	-	-	-	L34-R35	-	-
-	R35-K36	-	-	R35-K36	R35-K36	-	R35-K36	-
-	-	-	-	-	-	-	K36-G37	-
-	-	-	-	-	S40-E41	S40-E41		-
-	-	-	-	-		-	R42-V43	-
-	V49-Y50	-	-	-	V49-Y50 **	-		-
L51-A52	-	-	-	-	L51-A52	L51-A52	L51-A52	-
-	-	-	-	-	-	A52-A53	-	-
-	-	-	-	-	A53-V54	A53-V54	-	-
-	-	-	-	-	E56-Y57		-	E56-Y57
-	-	-	-	-	-		A60-E61	
-	-	-	-	-	L65-A66	L65-A66	L65-A66	
-	-	-	-	-	-	-	-	A70-R71
-	-	-	-	-	-	R71-D72	R71-D72	R71-D72
-	-	-	-	-	D72-N73	-	-	D72-N73
N73-K74	-	-	-	-	N73-K74	-	N73-K74	-
-	-	-	K74-K75	K74-K75	-	K74-K75	K74-K75	K74-K75
-	K75-T76	K75-T76	K75-T76	K75-T76	-	K75-T76	K75-T76	-
R77-I78	-	R77-I78	R77-I78	-	R77-I78	R77-I78	R77-I78	R77-I78
R81-H82	R81-H82	R81-H82		R81-H82		R81-H82	R81-H82	R81-H82
H82-L83	-	-	-	-	H82-L83	-	-	H82-L83
-	-	-	-	-	L83-Q84	-	-	L83-Q84
Q84-L85	-	-	-	-	Q84-L85	Q84-L85	Q84-L85	Q84-L85
						L85-A86		
A86-I87	-	-	-	-	-	A86-I87	-	A86-I87
I87-R88	I89-R90	I87-R88	I87-R88	I87-R88	I87-R88	I87-R88	I87-R88	I87-R88
-	-	-	-	-	-	-	K95-L96	-
-	-	-	-	-	-	-	L97-G98	-
-	-	-	-	-	-	T101-I102	T101-I102	-
-	-	-	I102-A203	-	I102-A203	-	-	-
-	A103-Q104	-	-	-	-	-	-	-
						I11-Q112		
-	-	-	-	-	-	-	-	L116-P117
V114-L115	V114-L115	V114-L115	V114-L115	V114-L115		-	-	-
-	-	-	-	-	-	L116-P117	-	-
-	-	E121-S122	-	-	-	-	-	-
A126-K127	A126-K127	A126-K127	A126-K127	A126-K127	A126-K127	-	A126-K127	A126-K127

* The molecular weights of the histone hydrolysis products were used to determine the corresponding sites of the hydrolysis based on a set of data from 8–10 spectra. ** Major sites of the hydrolysis are shown in red, moderate sites in black, and minor sites in green. Missing splitting sites are marked with a dash (-).

**Table 4 ijms-24-08636-t004:** Sites of H2A histone hydrolysis by IgGs against five individual histones 20 and 60 days after mice immunization with DNA-histones complex *.

Sites of H2A Hydrolysis after 20 Days after Mice Immunization with DNA-Histones Complex									
DNA20-aH1	DNA20-aH2A	DNA20-aH2B	DNA20-aH3	DNA20-aH4	DNA60-aH1	DNA60-aH2A	DNA60-aH2B	DNA60-aH3	DNA60-aH4
11 Sites	27 Sites	19 Sites	11 Sites	8 Sites	11 Sites	22 Sites	35 Sites	23 Sites	24 Sites
K9-A10 **	-	-	-	K9-A10	-	K9-A10	K9-A10	K9-A10	K9-A10
R11-A12	-	-	-	-	R11-A12	-	-	R11-A12	R11-A12
K13-A14	-	K13-A14	-	K13-A14	K13-A14	-	-	-	K13-A14
R17-S18 **	-	R17-S18	-	R17-S18	R17-S18	R17-S18	R17-S18	-	R17-S18
R20-A21	-	R20-A21	-	R20-A21	R20-A21	-	-	-	R20-A21
-	-	-	-	-	-	-	A21-G22	-	-
-	-	-	-	-	-	-	H31-R32	-	-
-	R32-L33 **	R32-L33	-	R32-L33	-	R32-L33	R32-L33	-	-
-	-	-	-	-	-	-	-	-	L34-R35
-	-	R35-K36	-	R35-K36	R35-K36	-	R35-K36	-	R35-K36
-	G37-N38	-	-	-	-	-	-	-	-
-	-	-	-	-	-	Y39-S40	Y39-S40	-	-
-	S40-E41	-	-	-	-	-	-	S40-E41	-
-	E41-R42	-	-	E41-R42	-	-	E41-R42	-	E41-R42
-	-	R42-V43	-	-	-	-	-	-	-
-	V43-G44	-	-	-	-	-	-	-	-
-	-	-	A45-G46	-	-	-	-	-	-
-	-	-	V49-Y50	-	-	-	-	V49-Y50	-
-	Y50-L51	-	-	-	-	Y50-L51	Y50-L51	Y50-L51	-
-	-	L51-A52	L51-A52	L51-A52	-	L51-A52	L51-A52	L51-A52	L51-A52
-	A53-V54	A53-V54		A53-V54	-	-	A53-V54	-	-
-	-	-	-	-	-	-	-	-	V54-L55
-	-	-	-	-	-	L55-E56	L55-E56	-	-
-	-	-	-	E56-Y57	-	-	-	E56-Y57	-
-	Y57-L58	-	-	-	-	-	-	Y57-L58	-
-	L58-T59	-	L58-T59	-	-	L58-T59	-	-	-
-	-	-	-	-	-		-	T59-A60	-
-	-	-	-	-	-	A60-E61	-	-	-
-	-	-	L63-E64	-	-	L63-E64	-	-	-
-	-	-	E64-L65	-	-	E64-L65	E64-L65	E64-L65	-
-	L65-A66	-	-	-	-	-	-	-	-
-	G67-N68	-	-	-	-	-	-	G67-N68	-
-	-	-	-	-	-	-	-	N68-A69	-
-	-	-	-	-	-	-	A70-R71	-	-
-	-	-	-	-	-	R71-D72	-	-	-
-	D72-N73	-	-	D72-N73	-	D72-N73	-	-	-
-	-	-	-	-	-	-	N73-K74	N73-K74	-
-		K74-K75	-	K74-K75	K74-K75	-	K74-K75	-	K74-K75
K75-T76	K75-T76	K75-T76	K75-T76	K75-T76	K75-T76	-	-	K75-T76	-
-	-	-	-	T76-R77	-	-	T76-R77	-	T76-R77
-	R77-I78	R77-I78	R77-I78	R77-I78	R77-I78	R77-I78	R77-I78	-	R77-I78
-	-	-	-	-	-	-	-	I78-I79	I78-I79
-	-	-	-	-	-	I79-P80	-	-	-
R81-H82	R81-H82	R81-H82	-	R81-H82		R81-H82	R81-H82	-	R81-H82
-	H82-L83	-	-	-	-	-	H82-L83	H82-L83	-
-	-	-	-	-	-	-	-	L83-Q84	-
-	Q84-L85	Q84-L85	-	-	-	Q84-L85	Q84-L85	Q84-L85	Q84-L85
-	L85-A86	L85-A86	-	L85-A86	-	-	-	-	L85-A86
-	-	-	-	A86-I89	-	-	-	-	-
I87-R88	I87-R88	-	-	I87-R88	I86-R88	I87-R88	I87-R88	-	I87-R88
-	-	-	-	-	-	-	-	-	-
-	L93-N94	-	-	-	-	-	-	-	-
-	-	K95-L96	-	K95-L96	-	-	K95-L96	-	K95-L96
-	-	L96-L97	-	L96-L97	-	-	L96-L97	-	
-	-	-	-	-	-	R99-V100	-	-	R99-V100
-	T101-I102	-	-	-	-	T101-I102	T101-I102	T101-I102	
-		-	-	-	-	-	-	-	I102-A103
A103-Q104	A103-Q104	-	-	-	-	-	A103-Q104	A103-Q104	
-	Q104-G105	Q104-G105	Q104-G105	Q104-G105	-	-	-	Q104-G105	Q104-G105
-	-	-	-	-	-	G105-G106	-	-	-
-	-	-	-	-	-	-	G106-V107	-	-
-	-	-	V107-K108	-	-	-	-	-	-
-	N110-I111	-	N110-I111	-	-	-	-	-	-
-	-	-	-	-	-	I111-Q112	I111-Q112	I111-Q112	-
Q112-A113	-	-	-	-	-	-	Q112-A113	-	-
-	A113-V144	-	-	-		-	A113-V114	A113-V114	-
-	-	V114-L115	-	-	V114-L115	-	-	-	-
-	-	-	-	-	-	-	L116-V117	-	-
-	-	-	-	-	-	-	T120-S121	-	-
-	E121-S122	-	-	-	-	-		-	-
-	-	-	-	-	-	-	S122-E121	-	-
-	H123-H124	-	-	-	-	-	H123-H124	-	-
A126-K127	-	A126-K127	-	A126-K127	A126-K127	-	A126-K127	-	A126-K127

* The molecular weights of the histone hydrolysis products were used to determine the corresponding sites of the hydrolysis based on a set of data from 8–10 spectra. ** Major sites of hydrolysis are shown in red, moderate sites in black, and minor sites in green. Missing splitting sites are marked with a dash (-).

## Data Availability

The data that supports the results of this study are included in the article and its Appendix A.

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
