# Peer review of "EAE of Mice: Enzymatic Cross Site-Specific Hydrolysis of H2A Histone by IgGs against H2A, H1, H2B, H3, and H4 Histones and Myelin Basic Protein"

_ijms, 2023, doi:10.3390/ijms24108636_

Round 1

Reviewer 1 Report

It's a very interesting topic with a well-written and designed study. However, the authors need to do some revisions and improve some points to make the paper worthy of publication: 

- Introduction: too long and the aims of the study are not highlighted. 

- The method section is lacking and its content is mixed with the introduction. In this paper, authors jumped from Introduction to results; maybe they might have omitted the title and mixed both contents. 

- The design of the study can be improved with a clearer description. 

- English style requires minors modifications. 

Author Response

Comments and Suggestions for Authors

It's a very interesting topic with a well-written and designed study. However, the authors need to do some revisions and improve some points to make the paper worthy of publication: 

- Introduction: too long and the aims of the study are not highlighted. 

Answer:

Sorry, this is not our first article on the description of abzymes. In general, there are not many publications on natural abzymes. This is, in a sense, a direction in science that is far from the understanding of many immunologists. With this in mind, when writing the introduction, we tried to reflect all aspects of the abzimology associated with the article, in connection with which the reviewers most often have questions.

Sorry, but from our point of view, the purpose of this study is indicated in the introduction.

The text from Introduction: It seemed interesting to what extent the phenomenon of enzymatic cross-reactivity between Abzs against histones and MBP is common for humans and animals with different AIDs. In addition, it was important to understand whether there is an unusual enzymatic cross-reactivity of Abs-Abzs against histones and MBP only in the case of H4 or also for other histones. In addition, the analysis of abzymes corresponding to different stages of EAE development by C57BL/6 mice allows an understanding of how the relative activities of abzymes and their substrate specificity concerning individual histones and MBP can change depending on the stage of pathology.

Here, it showed that abzymes of mice against every of five individual histones and MBP possess catalytic cross-reactivity in the splitting of H2A histone. Moreover, it was demonstrated that abzymes against each of the five histones and MBP corresponding to different stages of EAE development can hydrolyze H2A histone with different efficiency and in various specific sites.

Sorry, we somehow do not understand what else could be added with the designation of the purpose of the study

- The method section is lacking and its content is mixed with the introduction. In this paper, authors jumped from Introduction to results; maybe they might have omitted the title and mixed both contents. 

Answer:

Sorry, but we have prepared the article in accordance with the requirements of the Journal. In this Journal, after the Introduction, there should be paragraphs of Results, then Discussion with following paragraph Materials and methods and finally Conclusions.

- The design of the study can be improved with a clearer description. 

Answer:

Sorry, it's not entirely clear what you mean, because the design of the article was made in accordance with all the rules of the journal

- English style requires minors modifications. 

Answer:

English style was corrected

Thank you very much for your comments.

Best wishes

Professor Georgy Nevinsky

Reviewer 2 Report

The article by Andrey E. Urusov et. al. entitled " EAE of mice: enzymatic cross site-specific hydrolysis of H2A histone by IgGs against H2A, H1, H2B, H3, and H4 histones and myelin basic protein " is a fascinating study on systematic analysis of the number and type of histone H2A cleavage sites by IgG-abzymes.

 The authors found that IgG-abzymes against MBP and five individual histones showed unusual polyreactivity in complex formation and enzymatic cross-reactivity in the specific hydrolysis of H2A histone.

Mass spectrometry was used to determine the number and the hydrolysis sites of H2A by IgG-abzymes from different stages of EAE evolution and against individual histones and MBP. Then, possible reasons for the catalytic cross-reactivity and great differences in the number and type of histone H2A cleavage sites were analyzed.

In general, the article is written clearly and presents interesting data, even if the figures and tables are not well represented and there are several typo and grammatical error in English. This article would be of interest to scientists who are focusing on the study of catalytic antibody and autoimmune disease.

I have the following comments and concern.

1.      Was cleavage of histones other than H2A observed? Please describe whether IgG-Abzymes hydrolyze histones other than H2A.

2.      Why did the number of H2A hydrolysis sites increase during the spontaneous development of EAE?

3.      The results of MALDI mass spectrometry are difficult to see. Please improve size and resolution.

Minor points

There are many careless typographical errors throughout the manuscript, even in the figure legends. I would recommend the authors to proofread the manuscript carefully. Below are some examples.

Page 7, Figure1A and 1B, “H2B” on the left side of picture should be “H2A”.

Page 8, line 329, “specters” should be “spectra”, Please correct.

Page 9 and 10, Figure 3-6, Please write IgG concentration as Figure 1.

Page 11-13, Figure 7-9, 11 and 12, In legend “Major site of H1” should be “Major site of H2A”

Page 26, line 863, “3-mice-old” should be “3-month-old”, Please correct.

Author Response

The article by Andrey E. Urusov et. al. entitled " EAE of mice: enzymatic cross site-specific hydrolysis of H2A histone by IgGs against H2A, H1, H2B, H3, and H4 histones and myelin basic protein " is a fascinating study on systematic analysis of the number and type of histone H2A cleavage sites by IgG-abzymes.

 The authors found that IgG-abzymes against MBP and five individual histones showed unusual polyreactivity in complex formation and enzymatic cross-reactivity in the specific hydrolysis of H2A histone.

Mass spectrometry was used to determine the number and the hydrolysis sites of H2A by IgG-abzymes from different stages of EAE evolution and against individual histones and MBP. Then, possible reasons for the catalytic cross-reactivity and great differences in the number and type of histone H2A cleavage sites were analyzed.

In general, the article is written clearly and presents interesting data, even if the figures and tables are not well represented and there are several typo and grammatical error in English. This article would be of interest to scientists who are focusing on the study of catalytic antibody and autoimmune disease.

I have the following comments and concern.

  1. Was cleavage of histones other than H2A observed? Please describe whether IgG-Abzymes hydrolyze histones other than H2A.

Answer:

Yes, all antibodies against all five histones hydrolyze any of the five histones and MBP. However, there is as much data and information on the hydrolysis of each of the hislones as  for H2A histone. These results can be published only later in separate articles. 

  1. Why did the number of H2A hydrolysis sites increase during the spontaneous development of EAE?

Answer:

In fact, the number of hydrolysis sites for five different histones versus five histones can either increase strongly or decrease markedly during the spontaneous development of EAE. It depends on the histone and on specific antibodies against different histones.

The exact answer to the question why the spontaneous development of EPE is accompanied by an increase in the sites of ppp hydrolysis by antibodies against different histones is currently quite difficult. First, MS of people is at least a two-phase autoimmune disease [74]. The cascade of many reactions at the first inflammatory phase is very sophisticated, involving many cytokines, chemokines, enzymes, proteins, and other compounds inducing macrophages and other cells producing NO radicals and osteopathin. Therefore, during the analysis of immunological, biochemical,  and clinical indices of MS, every current phase of the disease must be considered, including changing in immunoregulation, exhaustion of different compensatory and adaptive mechanisms, and systemic metabolic changes [74]. Since spontaneous EAE in mice proceeds in several stages, on different stages, autoantigens, including histones, and MBP, can form complexes with a variety of proteins, nucleic acids, lipids, polysaccharides, cells, etc. This may lead to an expansion of the number and repertoire of antibodies capable of hydrolyzing H2A

  1. The results of MALDI mass spectrometry are difficult to see. Please improve size and resolution.

Answer:

I agree with you, but there are many drawings and they are large. Considering this, if you make it bigger, then there will be a lot of empty spaces in the text, since the figure must be given immediately after it is mentioned. However, taking into account your comments, we have increased the size of the drawings.

Minor points

There are many careless typographical errors throughout the manuscript, even in the figure legends. I would recommend the authors to proofread the manuscript carefully. Below are some examples.

Page 7, Figure1A and 1B, “H2B” on the left side of picture should be “H2A”.

Answer:

It was corrected

Page 8, line 329, “specters” should be “spectra”, Please correct.

Answer:

It was corrected

Page 9 and 10, Figure 3-6, Please write IgG concentration as Figure 1.

Answer:

It was corrected

Page 11-13, Figure 7-9, 11 and 12, In legend “Major site of H1” should be “Major site of H2A”

Answer:

It was corrected

Page 26, line 863, “3-mice-old” should be “3-month-old”, Please correct.

Answer:

It was corrected

Thank you very much for your valuable comments.

Best wishes

Professor Georgy Nevinsky

Reviewer 3 Report

I have no questions with current version of the article.

Author Response

I have no questions with current version of the article.

Answer:

Tjhanks for very short comments

Sincerely

Prof. Georgy A. Nevinsky

Round 2

Reviewer 1 Report

No more comments